# Homeostatic plasticity in the retina is associated with maintenance of night vision during retinal degenerative disease

**Henri Leinonen[1†]\*, Nguyen C Pham[2†], Taylor Boyd[2], Johanes Santoso[1], Krzysztof Palczewski[1,3], Frans Vinberg[2]\***

[1]Gavin Herbert Eye Institute, Department of Ophthalmology, University of California, Irvine, Irvine, United States; [2]John A. Moran Eye Center, Department of Ophthalmology and Visual Sciences, University of Utah, Salt Lake City, United States; [3]Departments of Physiology and Biophysics, and Chemistry, University of California, Irvine, Irvine, United States

**Abstract** Neuronal plasticity of the inner retina has been observed in response to photoreceptor degeneration. Typically, this phenomenon has been considered maladaptive and may preclude vision restoration in the blind. However, several recent studies utilizing triggered photoreceptor ablation have shown adaptive responses in bipolar cells expected to support normal vision. Whether such homeostatic plasticity occurs during progressive photoreceptor degenerative disease to help maintain normal visual behavior is unknown. We addressed this issue in an established mouse model of Retinitis Pigmentosa caused by the P23H mutation in rhodopsin. We show robust modulation of the retinal transcriptomic network, reminiscent of the neurodevelopmental state, and potentiation of rod – rod bipolar cell signaling following rod photoreceptor degeneration. Additionally, we found highly sensitive night vision in P23H mice even when more than half of the rod photoreceptors were lost. These results suggest retinal adaptation leading to persistent visual function during photoreceptor degenerative disease.

**\*For correspondence:**
hleinone@uci.edu (HL);
frans.vinberg@utah.edu (FV)

[†]These authors contributed equally to this work

**Competing interests:** The authors declare that no competing interests exist.

## Introduction

Neurons and neural networks require mechanisms for maintaining their stability in the face of numerous perturbations occurring over a lifetime. Homeostatic plasticity is the process whereby the activity of the neuron or neural network is maintained (*Burrone, 2003*; *Turrigiano, 2012*). Homeostatic plasticity is a relatively new concept first described in the central nervous system in 1998 (*Turrigiano et al., 1998*). Since then various homeostatic plasticity mechanisms have been reported in different regions of the brain, including the hippocampus and somatosensory and visual cortices, using in vivo models and ex vivo and in vitro preparations (*Gainey and Feldman, 2017*; *Turrigiano, 2012*; *Turrigiano, 2017*). As examples, binocular visual deprivation triggers compensatory synaptic changes in the primary visual cortex leading to the stabilization of spiking activity (*Desai et al., 2002*; *Goel et al., 2006*; *Goel and Lee, 2007*), and stroke can trigger homeostatic plasticity that is thought to compensate for neural damage and play an important role in early phase rehabilitation (*Murphy and Corbett, 2009*). Homeostatic plasticity counteracts insufficient activity in neural synapses (*Burrone, 2003*; *Turrigiano, 2012*), but it also works to prevent saturation of synapses that could be caused by positive feedback mediated by Hebbian plasticity, which is better known for its role in experience-dependent plasticity, learning and memory (*Abraham and Bear, 1996*).

The neural retina of the eye is considered relatively stable both structurally and functionally after postnatal development. Still, remodeling of the retina in response to photoreceptor degenerative diseases is well established. Retinal remodeling often has been described as a maladaptive process

that can exacerbate the disease and impede treatment strategies designed to rescue light responsiveness of the retina (*Fariss et al., 2000*; *Jones and Marc, 2005*; *Marc and Jones, 2003b*; *Telias et al., 2019*; *Toychiev et al., 2013*). It involves formation of ectopic synapses and expression of disruptive spontaneous oscillatory activity in the post-receptoral inner retina during photoreceptor degenerative disease (*Goo et al., 2011*; *Haverkamp et al., 2006*; *Margolis et al., 2008*; *Michalakis et al., 2013*; *Stasheff, 2008*; *Toychiev et al., 2013*; *Tu et al., 2015*). In contrast, recent evidence supports adaptive rather than solely destructive neural changes. For example, dendritic compensatory synapse formation occurs in retinal bipolar cells when various retinal cells, including rod and cone photoreceptors, have been ablated (*Beier et al., 2017*; *Beier et al., 2018*; *Care et al., 2019*; *Johnson et al., 2017*; *Shen et al., 2020*). These studies suggested that rod or cone bipolar cell dendrites can extend their arbors and form new synapses with their correct targets (i.e. rod and cone photoreceptor cells) either within their normal territory or sometimes outside of their normal dendritic field to compensate for the reduced number of photoreceptor cells. On the other hand, the most recent of such studies showed functional compensation without any observable anatomical changes in rod bipolar cells even after ablation of ~50% of rods in adult mice (*Care et al., 2020*). These homeostatic changes could help maintain retinal output and vision at or near normal levels. It is likely that the mechanisms, onset and time course of degeneration as well as the cell types in question all contribute to remodeling in the retina, and whether it will have a supportive or deleterious effect on vision. Thus, it will be critical to determine the nature of remodeling and plasticity under various conditions of retinal degeneration.

Whether homeostatic plasticity is present during a progressive retinal degenerative disease is unknown. We hypothesized that an inherited photoreceptor degenerative disease that progresses in a moderate manner would promote homeostatic plasticity supporting normal vision. We tested this hypothesis using an established mouse model of the most common form of Retinitis Pigmentosa (RP) caused by a dominant P23H mutation in the rod visual pigment, rhodopsin (*Sakami et al., 2014*; *Sakami et al., 2011*). By conducting whole retinal RNA-sequencing, in vivo and ex vivo electrophysiology, as well as behavioral experiments in diseased and control mice during the course of disease progression, we demonstrate that rod photoreceptor degeneration triggers homeostatic plasticity and the potentiation of rod–rod bipolar cell signaling that is associated with the maintenance of highly sensitive scotopic vision. This type of compensatory mechanism may explain why patients with inherited retinal diseases can maintain normal vision until a relatively advanced disease state is reached and could inspire novel treatment strategies for blinding diseases.

## Results

### Progression of retinal degeneration and functional decline in heterozygous P23H mice

Retinal remodeling following photoreceptor cell death has been observed in a number of genetic and induced animal models of retinal degeneration (*Beltran, 2009*; *Chang et al., 2007a*; *LaVail et al., 2018*; *Petersen-Jones, 1998*; *Ross et al., 2012*), as well as in postmortem human eye specimens from patients with retinitis pigmentosa (RP) or age-related macular degeneration (AMD) (*Fariss et al., 2000*; *Jones et al., 2016a*; *Jones et al., 2016b*; *Li et al., 1995*). In the current study, we utilized a well-established mouse model of autosomal dominant RP (*Sakami et al., 2014*; *Sakami et al., 2011*), caused by a heterozygous P23H mutation in the rhodopsin gene ($Rho^{P23H/WT}$). These mice will be referred to in this report as P23H mice. We started by confirming the suitability of this model for our study, focusing on rod-mediated retinal signaling and night vision. We followed structural changes of the retina by standard histology and optical coherence tomography (OCT) imaging, and confirmed that the outer nuclear layer (ONL) where the photoreceptor nuclei reside, and the inner and outer segments (IS and OS, respectively) of rod and cone photoreceptors progressively degenerated, while inner retinal layers appeared to remain anatomically intact (*Figure 1A–G*). In mice, the thickness of the ONL correlates with the number of surviving rod photoreceptors. We found 20%, 60% and 73% decreases in central retinal ONL thickness at 1-, 3- and 5 months of age, respectively, in P23H mice compared to WT littermates (*Figure 1H*). Next, using in vivo electroretinography (ERG) we tested the light-activated mass electrical response arising from the retina. We analyzed the a-wave of the fully dark-adapted retina, which is primarily a rod-dominant response

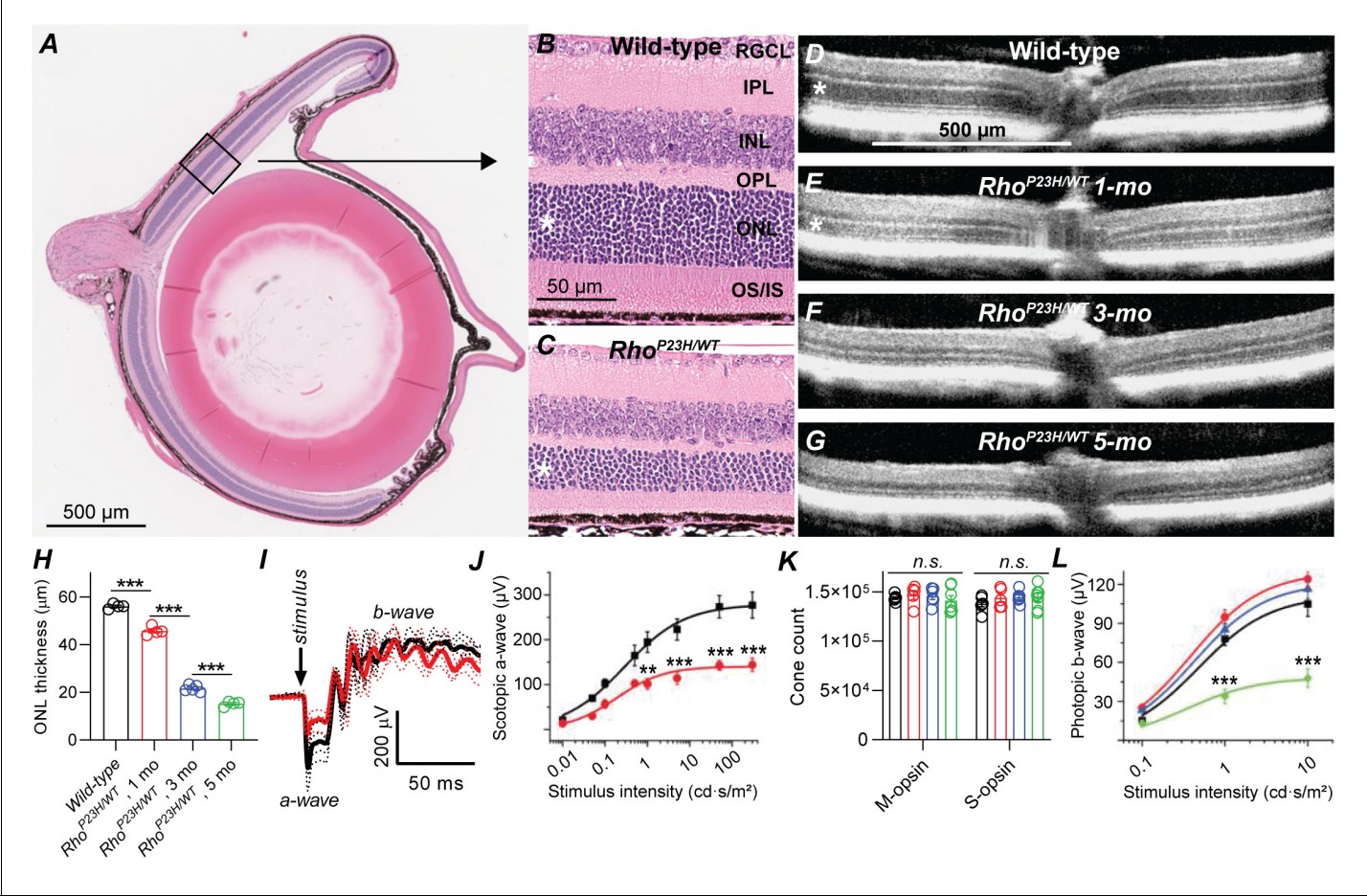

**Figure 1.** Characterization of retinal degeneration progression in heterozygote P23H rhodopsin-mutated mice. (**A**) Representative histology image of a wild-type (WT) mouse eye. (**B–C**) Representative magnified histology images of 1-month-old WT (**B**) and P23H (**C**) dorsal retinas. Asterisks mark the ONL, i.e. photoreceptor nuclei layer. RGCL, retinal ganglion cell layer; IPL, inner plexiform layer; INL, inner nuclear layer; OPL, outer plexiform layer (the site of photoreceptor-bipolar cell synapse); IS/OS, photoreceptor inner and outer segments. (**D–G**) Representative optical coherence tomography images. (**H**) ONL thickness analysis from central retina (WT, black, n = 4; P23H one mo, red, n = 4; P23H three mo, blue, n = 5; P23H five mo, green, n = 4 mice). (**I**) Scotopic ERG waveforms in response to 100 cd•s/m2 flash in 1-month-old WT (black) and P23H (red) mice (WT, black, n = 4 (8); P23H, red, n = 4 (8) mice(eyes)). (**J**) ERG a-wave amplitude analysis at 1 month of age. (**K**) Total M- and S-opsin counts in whole mount retinas (WT, black, n = 6; P23H one mo, red, n = 5; three mo, blue, n = 6; five mo, green, n = 7 mice) (**L**) Photopic ERG b-wave amplitude analysis (WT, black, n = 6 (12); P23H one mo, red, n = 8 (16); three mo, blue, n = 7 (14); five mo, green, n = 5 (10) mice(eyes)). Bonferroni post hoc tests: *p<0.05, **p<0.01, ***p<0.001. The online version of this article includes the following source data and figure supplement(s) for figure 1:

**Source data 1.** Average central ONL thickness (H), scotopic ERG a-wave amplitude (J), whole retina M- and S-cone count (K) and photopic b-wave amplitude (L) for individual mice.

**Figure supplement 1.** Progressive outer retina degeneration in P23H mice.

with a minute contribution from cones. The dark-adapted a-wave showed an almost 50% suppression in P23H mice by 1 month of age (*Figure 1I,J*) consistent with an earlier study by *Sakami et al., 2011*.

Cone cells comprise only 3% of the photoreceptor population in pigmented mice. Therefore, the experiments just described lacked the necessary sensitivity to evaluate cone cell survival. To investigate the fate of cone cells in P23H mice, we used an immunohistochemical approach to determine the number of cone cells and light-adapted, cone-dominant ERG recordings to assess their functional status. The total count of short- and medium-wavelength sensitive S- and M-cones, respectively, remained constant even at 5 months of age, the oldest time point tested (*Figure 1K*).

ANOVA analysis revealed a modest but statistically significant ($F_{1,12}$=5.25, p=0.04) increase in the photopic ERG amplitude in P23H mice compared to WT littermates at 1 month of age, which continued at 3 months of age (*Figure 1L*). However, cone function started to decline thereafter, and at 5 months of age the photopic ERG amplitudes significantly decreased in P23H mice.

Next, using eyecup cryosections and antibody staining, we evaluated the integrity of the synapse site between photoreceptors and bipolar cells, the outer plexiform layer (OPL, *Figure 1—figure supplement 1*). We stained rod bipolar cells by anti-protein kinase C α (PKCα), horizontal cells by anti-calbindin, and photoreceptor synaptic terminals by anti-vesicular glutamate transporter 1 (VGLUT1) antibodies. PKCα and calbindin staining patterns and intensities appeared unchanged in 1-month-old P23H retinas compared to WT littermates. Instead, immunoreactivity of both these markers decreased with disease progression until 3- and even more until 5 months of age. VGLUT1 staining revealed slight thinning of the OPL already at 1 month of age in P23H mice, and this degeneration progressed with aging.

Finally, we performed anti-C-terminal-binding protein 2 (CtBP2) and anti-metabotropic glutamate receptor 6 (mGluR6) double-staining and optical sectioning fluorescence imaging with high magnification to show pre- and postsynaptic terminals, respectively, at the OPL in 1-month-old retinas (*Figure 2*). We observed intact morphology and density of the pre- and postsynaptic terminals, but the count of synapses was decreased by ~20% in P23H retinas. This is likely attributable to the overall thinning of the OPL.

These data demonstrate that rod photoreceptors in the P23H mouse model of RP are preferentially targeted in the early stage of disease and that the loss of these rod photoreceptors occurs at a moderate rate (*Figure 1*). The loss of rods is accompanied with a modest decrease of photoreceptor-to-bipolar cell synapse count but with no change in synaptic morphology at the light microscopy level (*Figure 2*). Therefore, the model is suitable for investigating the compensatory network changes that arise during progressive rod cell degeneration.

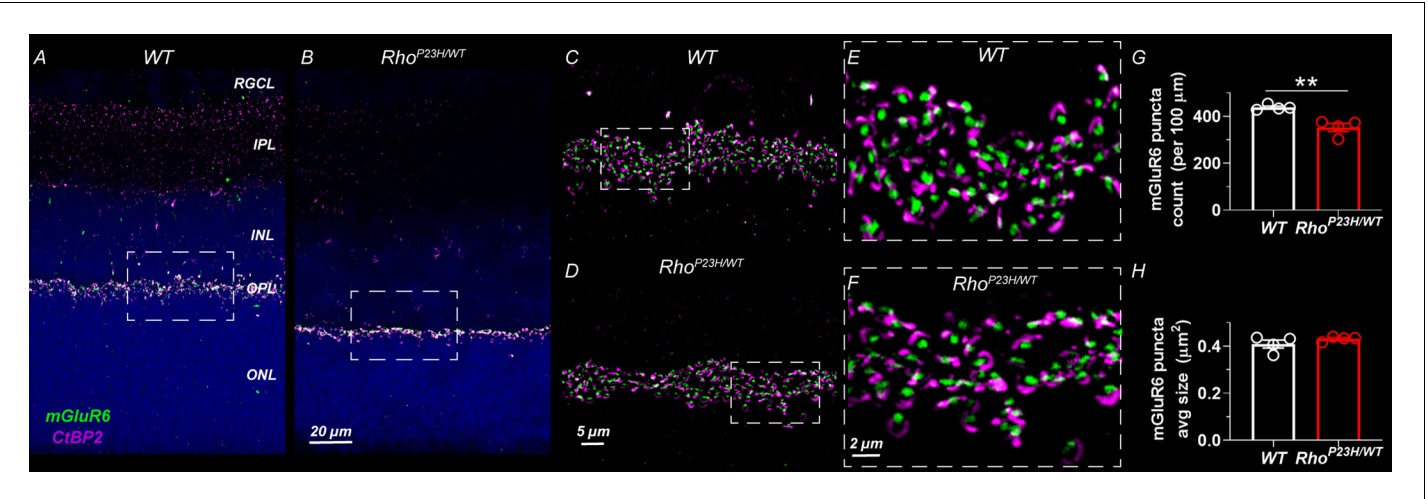

**Figure 2.** Decreased number of photoreceptor-bipolar cell synapses in 1-month-old P23H mouse retinas. (**A**) Representative sections of wild-type and (**B**) P23H mouse retinas stained with mGluR6 (green) and CtBP2 (magenta) antibodies that stain the postsynaptic and presynaptic compartments at the photoreceptor-bipolar cell synapse, respectively. Images A and B were acquired with 40x objective. Dashed line boxes show where images **C-D** acquired with 100x objective were taken. **E** and **F** are 3x digital zoom images from **C-D** (dashed line boxes show zoomed location). The photoreceptor-to-bipolar cell synapses show normal appearance in P23H retinas; however, the count is moderately decreased as shown in graph **G**. (**H**) Average mGluR6 puncta size is comparable in WT and P23H retinas. Four biological replicates and five technical replicates in each were inspected per group. mGluR6 puncta parameters were averaged of five technical replicates per sample. T-test: **p<0.01.

The online version of this article includes the following source data for figure 2:

**Source data 1.** Average mGluR6 puncta count per 100x30 µm$^2$ counting window (G) and average mGluR6 puncta size (H) for individual mice.

## RNA-sequencing from retinas reveals a robust neural network adaptation in early RP

We initially used RNA-sequencing (RNA-seq) to determine whether the transcriptomic profile of the retina changed in a manner that might provide functional adaptation in response to rod photoreceptor degeneration. Because at 1 month the P23H mouse retina has matured but has not yet degenerated severely, we believed this period would be optimum for RNA-seq analysis (*Figure 1H*). We found 2721 downregulated and 2683 upregulated genes in P23H mice compared to WT littermates (*Figure 3A*). There were few differentially expressed genes between sexes, unrelated to disease (*Supplementary files 1* and *2*), and therefore we pooled samples from both genders in the remainder of the analyses.

We used gene ontology analysis (GO) to objectively identify the 20 most downregulated and 20 most upregulated gene classes in P23H mouse retinas based on the p-values. As expected, numerous photoreceptor-specific gene classes were significantly downregulated in P23H mice, such as those related to ciliary structure, phototransduction, and photoreceptor cell differentiation (*Figure 3B*). In contrast, GO analysis revealed robust increases in gene clusters associated with neural growth and development, encompassing such cellular processes as synapse organization, postsynaptic specialization, glutamatergic synapse formation, axonogenesis, and cell-cell adhesion (*Figure 3D*). Furthermore, KEGG pathway analysis showed a total of 74 statistically upregulated pathways including those involved in cell adhesion, axon guidance, and glutamatergic synapse processing (*Figure 3C*). As expected, several pathways associated with cell stress were also upregulated, including the MAPK, NF-kappa B, and TNF signaling pathways. To verify the activation of at least one of these key pathways, the MAPK signaling pathway, ERK phosphorylation was analyzed in whole retina extracts and found to be enhanced in P23H mice (*Figure 3—figure supplement 1*).

We performed another RNA-seq analysis at a more advanced disease state, at 3 months of age, to provide transcriptome comparison to the early disease state (*Figure 3—figure supplement 2*). Notably, this dataset was collected with a smaller sample size than the 1-month-old dataset, likely contributing to fewer detected differentially expressed (DE) genes: 1556 downregulated and 1595 upregulated genes in P23H retinas compared to WT littermates. This analysis revealed largely similar findings as the analysis performed with 1-month-old samples; however, upregulations in cell stress-associated pathways were expectedly more pronounced at 3 months compared to the 1-month time point (*Figure 3—figure supplement 2*). Importantly, top 20 lists of upregulated 'molecular function' GO term pathways at 1 month and 3 months correspond to each other well, and both show prominent enrichments in ion-channel-activity-related gene clusters (*Figure 3—figure supplement 3*). Furthermore, analyses at both age points show an expected downregulation of photoreceptor-enriched genes due to progressive loss of rods but increases in several postsynaptic transcripts that encode crucial proteins for ON bipolar cell depolarization, such as mGluR6 and TRPM1 (*Figure 3—figure supplement 4*). This pattern of regulation was confirmed from 1-month-old samples using quantitative PCR (*Figure 3—figure supplement 5*). Collectively, the RNA-seq data show a pattern wherein neural development and growth-related pathways are the highest upregulated transcriptomic pathways in the early disease state, accompanied by persistent upregulations in ion channel activity-related pathways and crucial postsynaptic elements at the bipolar cells, all consistent with neural network adaptation following the initial sensory defect in the retina.

All filtered RNA-seq analysis results, including all differentially expressed genes between genotypes, as well as GO, KEGG and predicted reactome pathways that reached the statistical cut-off criterion are presented in *Supplementary files 3–18*. Raw data is freely available in the Gene Expression Omnibus (GEO) database (https://www.ncbi.nlm.nih.gov/geo/) with accession numbers GSE152474 (1-month-old samples) and GSE156533 (3-month-old samples).

## Increased sensitivity of rod bipolar cells to their rod input in P23H mice

To evaluate if transcriptomic network adaptation in P23H mice in response to rod degeneration is associated with changes in retinal light signaling, we first recorded light-evoked responses from isolated mouse retinas using ex vivo ERG (*Figure 4A*). Ex vivo ERG allows quantitative dissection of photoreceptor ($R_{PR}$) and ON bipolar cell ($R_{BC}$) responses by using blockers for the metabotropic glutamate receptor (mGluR) and potassium channels in Müller glia (*Bolnick et al., 1979*; *Vinberg et al., 2014*), see Materials and methods (*Figure 4B–D*). As expected, we found that the maximal

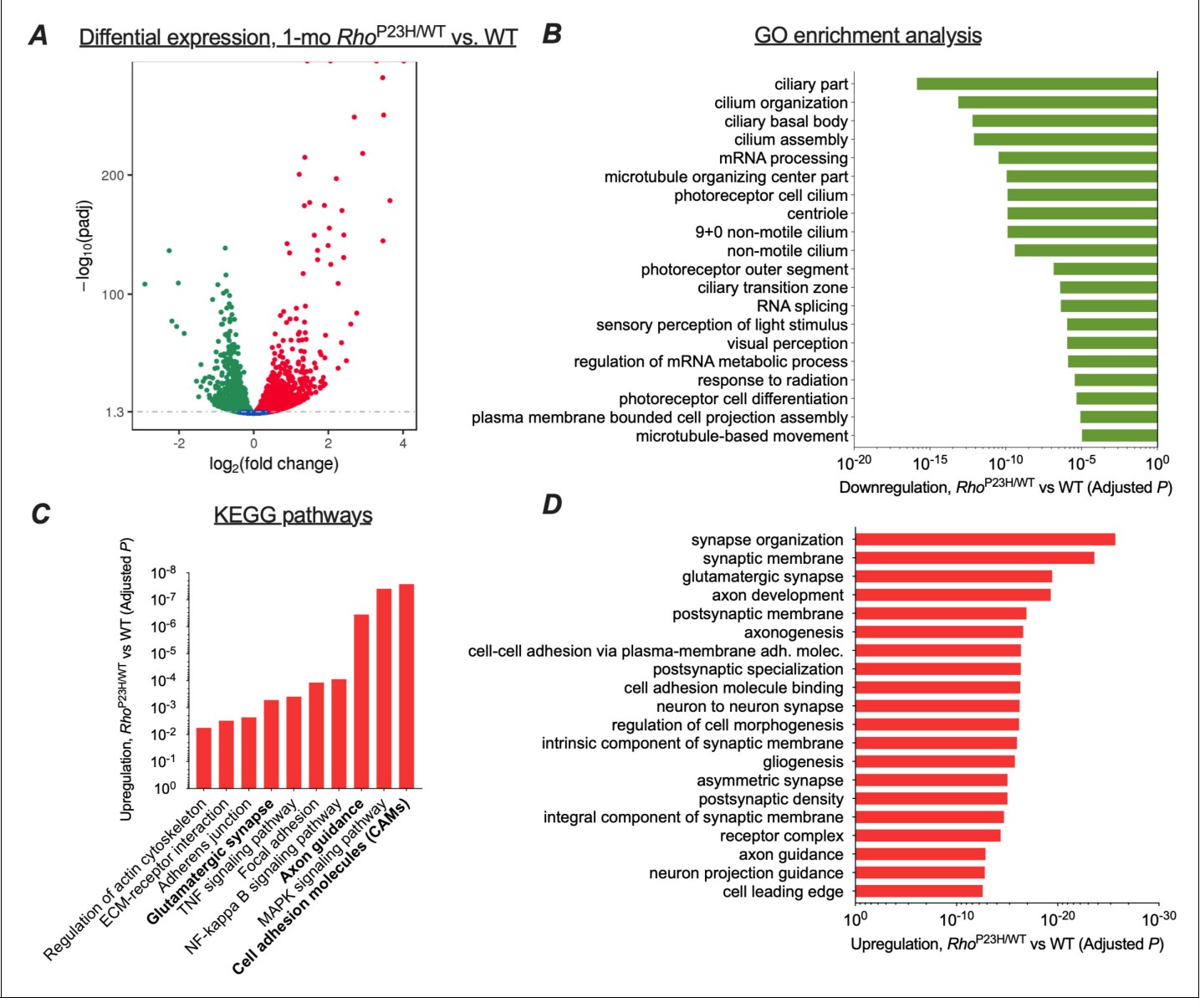

**Figure 3.** Transcriptomic analysis shows neural network adaptation to rod death in early stage retinitis pigmentosa. RNA-sequencing was performed from whole retinal extracts at 1 month of age in P23H mice (n = 7) and WT (n = 7) littermates. (**A**) A total of 5404 transcripts showed differential expression (downregulation, green; upregulation, red) between P23H and WT retinas. (**B**) Gene ontology (GO) term analysis shows expected downregulation in photoreceptor-dominant gene clusters. (**C**) KEGG analysis shows significant enrichments in cell/neuron adhesion and growth pathways, glutamatergic synapse formation and several pathways associated with both cell stress and synaptic plasticity. (**D**) GO term analysis illustrates the most significant upregulation in synaptic and neural development and growth pathways in P23H retinas.

The online version of this article includes the following source data and figure supplement(s) for figure 3:

**Figure supplement 1.** Activation of mitogen-activated protein kinase (MAPK) pathway in P23H retinas.

**Figure supplement 1—source data 1.** Immunoblot band intensities for a-tubulin, ERK1/2 and phospho-ERK1/2.

**Figure supplement 2.** Response to cell stress overwhelms retinal transcriptomic pathway regulation in intermediate stage retinitis pigmentosa.

**Figure supplement 3.** Top 20 list of GO molecular function terms is similar between 1- and 3-month-old retinas.

**Figure supplement 4.** P23H retinas show opposing and persistent mRNA expression regulation in genes required for synaptic transmission in pre- versus postsynaptic sites.

**Figure supplement 5.** A subset of important gene regulations was confirmed in 1-month-old retinas by qPCR.

**Figure supplement 5—source data 1.** Primary qPCR parameters for each technical and biological replicate.

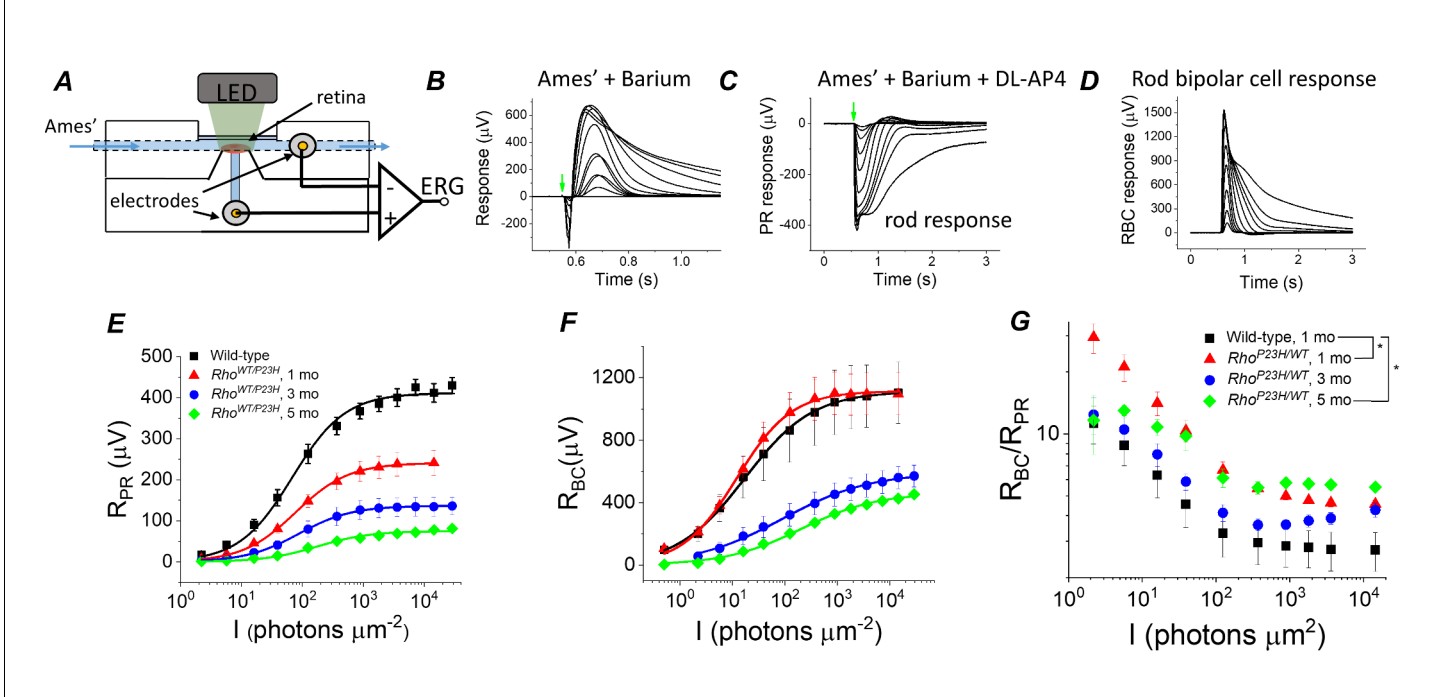

**Figure 4.** Sensitivity of bipolar cell responses to their photoreceptor input increases in P23H mice. (**A**) Custom ex vivo ERG specimen holder used to record transretinal voltage with two Ag/AgCl macro electrodes. Isolated retina without retinal pigment epithelium is mounted photoreceptor side up and perfused with Ames' medium. (**B**) Representative light flash responses from a dark-adapted WT control mouse retina perfused with Ames' containing 100 µM $BaCl_2$ to remove signal component arising from Müller cells. (**C**) Light responses from the same retina after addition of DL-AP4 eliminating metabotropic glutamatergic transmission to reveal the photoreceptor-specific responses. (**D**) Bipolar cell responses obtained by subtracting the traces shown in (**C**) from those in (**B**). Amplitude data for photoreceptor ($R_{PR}$, **E**) and bipolar ($R_{BC}$, **F**) responses, and $R_{BC}/R_{PR}$ ratio (**G**) as a function of light flash intensity in photons $µm^{-2}$ (505 nm). Mean ± SEM. In panel G, *p<0.05 (two-way ANOVA between-subjects effect). Smooth traces in E and F plot **Equation 1** fitted to the mean amplitude data. Statistics for $R_{max}$ and intensity required to generate half-maximal photoreceptor or ON bipolar cell response ($I_{1/2}$) are in **Table 1**. (Control, one mo, n = 4; P23H, one mo, n = 4; three mo, n = 3; five mo, n = 4 mice/retinas).

The online version of this article includes the following source data for figure 4:

**Source data 1.** Photoreceptor ($R_{PR}$, E) and bipolar cell ($R_{BC}$, F) amplitude, and $R_{BC}/R_{PR}$ ratio (G) data for individual mice.

photoreceptor response decreased by ~40% at 1 month of age in P23H mice as compared to control mice, and responses continued to decrease thereafter (**Figure 4E**, **Table 1**). These changes, initially at 1 month of age, were larger than expected when compared to the total loss of rod photoreceptors (**Figure 1H**), suggesting that the decrease in the light-sensitive current of the rods precedes rod degeneration in this model. Interestingly, however, light sensitivity measured as the light intensity to generate a half-maximal response ($I_{1/2}$) was not reduced in 1- to 3-month-old P23H mice (**Figure 4E**, **Table 1**). At 5 months of age, $I_{1/2}$ of P23H mice significantly increased (**Figure 4E**, **Table 1**). This may be due to the contribution from cone cells at this age (**Figure 1I and K**). Progressive reduction of rod photoreceptor light responses was expected and has been reported previously (**Sakami et al., 2011**).

However, in this study, the main issue was to determine whether the observed transcriptomic changes correlate with changes in how bipolar cells respond to their photoreceptor input in P23H mice. Strikingly, despite the ~40% decrease of photoreceptor input in 1-month-old P23H mice, their bipolar cell responses did not diminish at any of the light flash intensities utilized (**Figure 4F**). Since rod bipolar cells derive their responses by pooling input from several rod photoreceptors, we quantified the sensitivity of rod bipolar cells to their intrinsic input in two ways. First, we compared the ratio of bipolar cell and photoreceptor cell responses ($R_{BC}/R_{PR}$) between control and P23H mice at each light flash intensity (**Figure 4G**). This ratio increased in 1-month-old P23H mice compared to

**Table 1.** Ex vivo ERG parameters.

$I_{1/2}$, photons (505 nm) per flash/$\mu m^2$ from fitting *Equation 1* to photoreceptor (pharmacologically isolated) and bipolar cell (subtracted) amplitude data; $R_{max}$, maximum amplitude measured at plateau of the photoreceptor response and at the peak of the bipolar cell component in response to bright flash (~15,000 photons $\mu m^{-2}$); $R_{PR,1/2}$, photoreceptor response amplitude required for half-maximal bipolar cell response as determined by fitting *Equation 2* to $R_{PR}$ – $R_{BC}$ amplitude data. C57 background: Control, one mo, n = 4; P23H, one mo, n = 4; three mo, n = 3; five mo, n = 4 mice/retinas; *Gnat2*$^{-/-}$ background: Pooled control, 1–6 mo, n = 7 (11); P23H, one mo, n = 4 (6); five mo, n = 11(18); six mo, n = 2 (4) mice(retinas). *p<0.05, **p<0.005, ***p<0.001 in comparison to control data, two-tailed t-test. All data mean ± SEM.

| Genotype/age | Photoreceptors | | Bipolar cells | | |
|---|---|---|---|---|---|
| | I1/2 (hn/μm2) | Rmax (μV) | I1/2 (hn/μm2) | Rmax (μV) | RPR,1/2 (μV) |
| 1 month Control P23H/C57 | 74 ± 18 | 410 ± 20 | 24 ± 8 | 1,120 ± 203 | 76 ± 9 |
| 1 month P23H/C57 | 78 ± 10 | 242 ± 29** | 15 ± 2 | 1,152 ± 40 | 29 ± 6** |
| 3 month P23H/C57 | 98 ± 13 | 138 ± 24*** | 71 ± 30 | 547 ± 74* | 39 ± 1* |
| 5 month P23H/C57 | 285 ± 83* | 81 ± 7*** | 72 ± 7* | 388 ± 14* | 24 ± 1** |
| Pooled Control P23H/Gnat2 | 104 ± 17 | 545 ± 76 | 33 ± 12 | 830 ± 160 | 68 ± 9 |
| 1 month P23H/Gnat2 | 230 ± 40** | 238 ± 37** | 48 ± 6 | 670 ± 80 | 38 ± 6* |
| 5 month P23H/Gnat2 | 214 ± 32* | 176 ± 21*** | 42 ± 5 | 540 ± 50 | 28 ± 4*** |
| 6 month P23H/Gnat2 | 339 ± 47*** | 54 ± 8*** | 88 ± 9** | 220 ± 20** | 13 ± 2*** |

control mice, indicating a significant increase in the sensitivity of the bipolar cells. We also found less pronounced but statistically significant increases of $R_{BC}/R_{PR}$ in 5-month-old P23H mice (*Figure 4G*). These results are consistent with two prior studies showing less severe reduction of the in vivo ERG b-waves as compared to a-waves in a P23H transgenic rat model (*Aleman et al., 2001*; *Machida et al., 2000*). However, one caveat in analyzing $R_{BC}/R_{PR}$ is that the photoreceptor response amplitudes are not linear functions of light intensity. Thus, a second, more quantitative way to determine the sensitivity of bipolar cells to their photoreceptor input is to plot $R_{BC}$ as a function of $R_{PR}$ and determine the $R_{PR}$ required to generate a half-maximal $R_{BC}$ (=$RP_{P,1/2}$, see *Figure 5D–F* below and Materials and methods). This analysis revealed a significant decrease of $R_{P,1/2}$, i.e. sensitization of bipolar cells in P23H mice at all ages tested (1, 3 and 5 months) as compared to WT control mice (*Table 1*).

Since the mouse retina is rod-dominant, our results strongly suggest that rod bipolar cells become more sensitive to their rod input in P23H mice. However, at an older age, it is possible that the cone-mediated responses, which we found to be preserved longer compared to rods (*Figure 1J–L*), and even hyper-sensitized in the early disease state (*Figure 1L*), could dominate and explain the compensation of bipolar cell responses in P23H mice. To exclude this possibility and to study more quantitatively the implications of the P23H rhodopsin mutation in the rod signaling pathway, we crossed P23H mice with a cone transducin-α (*Gnat2*) knock-out (P23H/*Gnat2*$^{-/-}$) mouse line. The cones in these mice do not generate light responses but have no structural abnormalities (*Ronning et al., 2018*). We observed a qualitatively similar phenomenon in P23H/*Gnat2*$^{-/-}$ mice as in P23H knock-in mice (*Figures 4* and *5*). The maximal rod response amplitude was suppressed by more than half in 1-month-old P23H/*Gnat2*$^{-/-}$ mice compared to their littermate control mice (*Figure 5A*). A further decrease of the maximal rod response amplitudes in aging P23H/*Gnat2*$^{-/-}$ mice was accompanied by the desensitization of their rods as indicated by larger values of $I_{1/2}$ in 1–6 month-old P23H/*Gnat2*$^{-/-}$ mice compared to their littermate controls (*Figure 5A*, *Table 1*). The rod bipolar cell responses decreased less than the rod responses in 1- and 5-month-old P23H/*Gnat2*$^{-/-}$ mice before a relatively steep decrease from 5 to 6 months of age (*Figure 5B*). Since the decrease of $R_{BC}$ was slower compared to the decline in $R_{PR}$ with age, the $R_{BC}/R_{PR}$ ratios remained significantly higher up to 6 months of age in P23H/*Gnat2*$^{-/-}$ mice (the oldest age tested, *Figure 5C*). Moreover, we found an almost twofold decrease in photoreceptor input required to generate a half-maximal RBC response in 1-month-old P23H/*Gnat2*$^{-/-}$ mice as compared to that in control mice, and $R_{PR,1/2}$

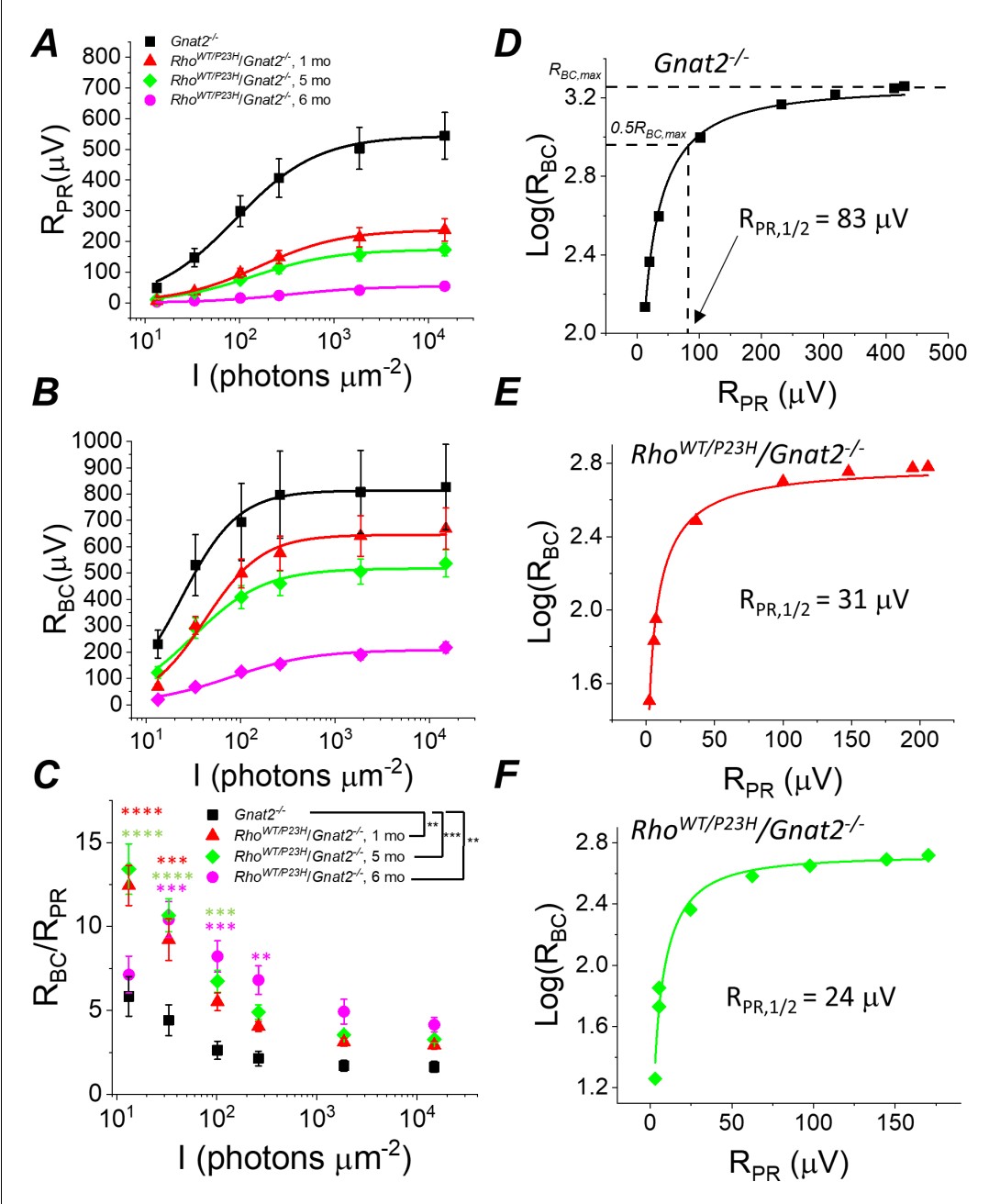

**Figure 5.** Sensitivity of rod bipolar cell responses to their rod input increases in P23H/$Gnat2^{-/-}$ mice. Ex vivo ERG rod (**A**) and RBC (**B**) response amplitudes, and their ratio (**C**) derived as in **Figure 4**. (**D–F**) RBC plotted as a function of $R_{PR}$ in individual retinas from $Gnat2^{-/-}$ control (**D**), and one month (**E**) and five month (**F**) P23H/$Gnat2^{-/-}$ mice. Smooth lines in A and B plot **Equation 1** fitted to mean data. Smooth lines in D-F plot **Equation 2** fitted to data from individual retinas. *p<0.05, **p<0.005, ***p<0.001, ****p<0.0001. Statistics for $R_{max}$ and intensity required to generate half-maximal rod or RBC response ($I_{1/2}$) as well as $R_{PR,1/2}$ are in **Table 1**. In C, two-way ANOVA followed by Bonferroni's post hoc test was used to compare between-subjects main effect and the intensity where a significant effect was found, respectively. (Pooled control, 1–6 mo, n = 7 (11); P23H, one mo, n = 4 (6); five mo, n = 11(18); six mo, n = 2 (4) mice(retinas)).

The online version of this article includes the following source data and figure supplement(s) for figure 5:

**Source data 1.** Photoreceptor ($R_{PR}$, A) and bipolar cell ($R_{BC}$, B) amplitude, and $R_{BC}/R_{PR}$ ratio (C) data for individual mice/retinas.

**Figure supplement 1.** Contribution of Müller cell response on the ex vivo ERG light responses in control and P23H mice.

**Figure supplement 1—source data 1.** Bipolar cell amplitudes in the absence (D) and presence (E) of Barium, and slow-PIII amplitudes (F) for individual mice/retinas.

continued to decrease down to ~20% of that in control retinas in 6-month-old P23H mice (*Figure 5D–F*, *Table 1*). Although Barium Chloride (BaCl$_2$) has been shown to be effective in blocking the ex vivo ERG component arising from Müller cells (*Bolnick et al., 1979*; *Nymark et al., 2005*; *Vinberg and Kefalov, 2015b*), the experiments above did not specifically assess the potential contribution of Müller cells to the b-wave amplitudes. To test that, we recorded ex vivo ERG responses from 1-month-old *Gnat2$^{-/-}$* and their littermate P23H/*Gnat2$^{-/-}$* mouse retinas perfused with Ames' medium without and with 100 µM BaCl$_2$ (*Figure 5—figure supplement 1A and B*). We found that b-wave amplitudes in the absence of BaCl$_2$ were smaller than those in the presence of barium (*Figure 5—figure supplement 1D,E*). This is probably because barium removes the negative slow PIII component that temporally overlaps with the positive b-wave. Comparison between 1-month-old control and P23H mice revealed that b-wave amplitudes were not affected by the P23H rhodopsin mutation in the absence or presence of barium (*Figure 5—figure supplement 1D,E*). Finally, we derived Müller cell responses by subtracting the responses recorded in the presence of barium from those recorded in the absence of barium from individual retinas across different light intensities (*Figure 5—figure supplement 1C,F*). Müller cell responses were not affected at dim light but tended to be somewhat smaller with the two brightest light intensities used in these experiments in the 1-month-old P23H mice as compared to those in control mice. These results do not support a role for increased Müller cell activity in explaining the observed potentiation of the b-wave responses in P23H mice.

Although the cones lacking GNAT2 do not generate light responses, light signal transmission from rods to the cone pathway through gap junctional coupling could be upregulated in P23H mice to compensate for the reduced rod input. To test this hypothesis, we determined the effect of a gap junction blocker, meclofenamic acid (MFA), on the photoreceptor a-wave and ON bipolar cell

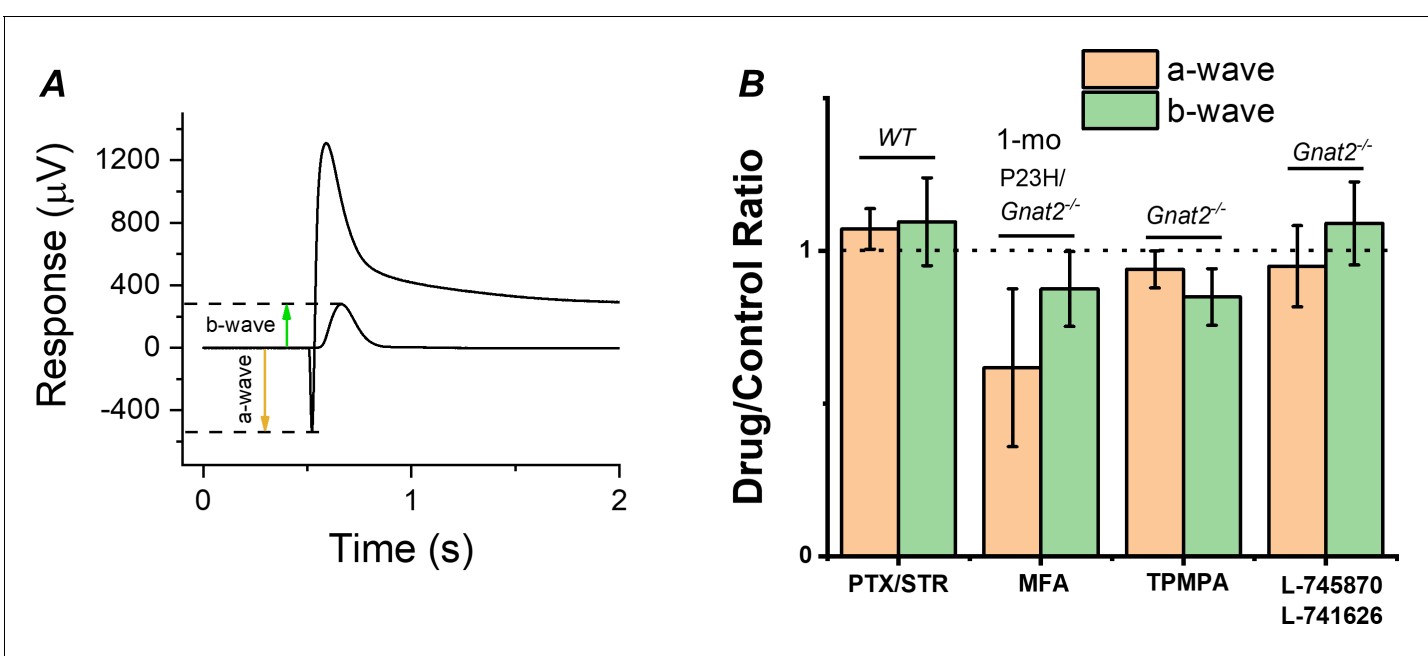

**Figure 6.** Contribution of gap junctions or inhibitory synaptic inputs to bipolar cells on the photoreceptor and bipolar cell components of the ex vivo ERG signal. (**A**) Example traces to dim and bright flashes recorded from a dark-adapted wild-type mouse retina perfused with Ames' containing 100 µM BaCl$_2$. B-wave amplitudes were measured from baseline using the dim flash responses and a-wave amplitudes from the baseline using the bright flash responses as indicated by arrows. (**B**) Mean ± SEM a- (brown, a-wave$_{drug}$/a-wave$_{control}$) and b-wave (green, b-wave$_{drug}$/b-wave$_{control}$) ratios determined from the each individual retina perfused with control solution and solution containing 100 µM of the gap junction blocker meclofenamic acid (MFA), 100 µM of picrotoxin (PTX) and 10 µM strychinine (STR), 10 µM of the dopamine D$_2$ and D$_4$ receptor antagonists L-745870 and L-741626, respectively, or 50 µM the GABA$_C$ receptor antagonist (1,2,5,6-Tetrahydropyridin-4-yl)methylphosphinic acid (TPMPA). Mouse genotypes for each drug experiment is indicated in Figure. No statistically significant changes in a- or b-wave amplitudes was observed with any of the drugs tested by paired t-test. n = 6 retinas for PTX/STR, n = 3 retinas for each MFA, TPMPA, and L-745870/L741626 experiments.

The online version of this article includes the following source data for figure 6:

**Source data 1.** Ratios of ex vivo ERG a-wave and b-wave amplitudes measured from individual retinas perfused in drug vs. control media (B).

b-wave by ex vivo ERG in 1-month-old P23H/$Gnat2^{-/-}$ mice (**Figure 6A,B**). If potentiation of the b-wave responses is due to increased signaling of rods to the cone pathway via gap junctions, we expected to see suppression of the b-wave amplitudes by MFA. Although MFA trended toward suppressing the b-wave amplitude, it appeared to cause even larger suppression of the a-wave amplitude (**Figure 6B**). This result suggests that electrical transmission from rods to cones or cone bipolar cells does not contribute significantly to the measured ex vivo ERG bipolar cell component in P23H/$Gnat2^{-/-}$ mice and cannot explain the observed potentiation of $R_{BC}$ (**Figure 5B–C**, **Table 1**). Another possibility for the increased sensitivity of the rod bipolar cells in P23H mice is reduced inhibitory input into rod bipolar cells, that is. disinhibition. To test this, we determined the contribution of GABA-, glycine-, and inhibitory dopaminergic signaling to the rod bipolar cell responses using ex vivo ERG in the retinas from $Gnat2^{-/-}$ and C57 wild-type mice (**Figure 6B**). No significant differences were found in the ERG a- or b-wave amplitudes between the control and treated conditions with any of the antagonists suggesting that disinhibition, as a mechanism, likely does not play a significant role in elevating rod bipolar cell responses observed in ex vivo ERG recordings from P23H mice (**Figure 5B**). Collectively, these results suggest that during rod degeneration and suppressed rod phototransduction triggered by the P23H rhodopsin mutation, the rod-rod bipolar cell signaling is strengthened via compensatory pathways that are most probably intrinsic to the rod axon terminal and/or rod bipolar cells.

## Rod-mediated visual contrast sensitivity remains relatively normal despite loss of majority of rods

Our molecular genetics and functional analyses showed that compensatory pathways in the retina are expected to support retinal output even after 50–70% degeneration or functional decline of the rods in P23H mice. To test if these compensatory changes can promote vision, we applied an optomotor response test (OMR; **Prusky et al., 2004**) to evaluate behavioral visual contrast sensitivity (CS) in P23H/$Gnat2^{-/-}$ mice and their $Gnat2^{-/-}$ littermate controls (**Figure 7A**). This reflexive test was selected to avoid confounding factors caused by potential plasticity in the cortical brain areas that could also promote vision even when retinal output is suppressed (**Koskela et al., 2020**). Our aim was to focus on the question of whether potentiation of the rod-rod bipolar cell signaling in P23H mice can help maintain high-sensitivity night vision. This goal was further facilitated by use of the $Gnat2^{-/-}$ background to abolish light responses of cones and the use of ambient light ranging from scotopic to photopic conditions. We found that CS decreased slightly in P23H/$Gnat2^{-/-}$ mice only at the dimmest background light up until 3 months of age in ambient light ranging from −3.3 to 0.85 log(Cd/m$^2$) (**Figure 7B,C**). At 5 months of age, CS was suppressed at the two dimmest ambient light levels (−3.3 and −2.4 log(Cd/m$^2$)), but not at brighter illuminances (**Figure 7D**). In contrast, a significant drop in performance occurred in 6-month-old P23H/$Gnat2^{-/-}$ mice at all except the brightest ambient light level (**Figure 7D**). A decrease in the sensitivity of rod-mediated vision was expected since absolute rod bipolar cell responses decreased in P23H mice as they aged beyond 1 month (**Figures 4F** and **5B**; **Sarria et al., 2015**). However, consistent with our finding that the rod bipolar cell sensitivity to their rod input was potentiated in P23H/$Gnat2^{-/-}$ mice up to 5 months of age, the decrease in scotopic contrast sensitivity was less severe than expected if based solely on the degeneration and functional decline of rod photoreceptors caused by the P23H rhodopsin mutation. It is possible that the removal of cone light responses in $Gnat2^{-/-}$ mice effects the remodeling process during rod degeneration caused by the P23H mutation in rhodopsin or some off-target effects that influence mouse behavior. To control for these possibilities, we also conducted the optomotor response tests in control and P23H mice on a C57Bl/6J background with functional cones. In these mice, we did not observe any change in contrast sensitivity in dim (scotopic), intermediate (mesopic) or bright (photopic) ambient light (**Figure 7—figure supplement 1**).

## Discussion

### P23H mice as a model to study the impact of early stage retinal remodeling on retinal function and vision

Retinal remodeling commonly refers to neuronal-glial changes in the retina in response to photoreceptor degeneration regardless of the etiology of the disease, and it is characterized by three

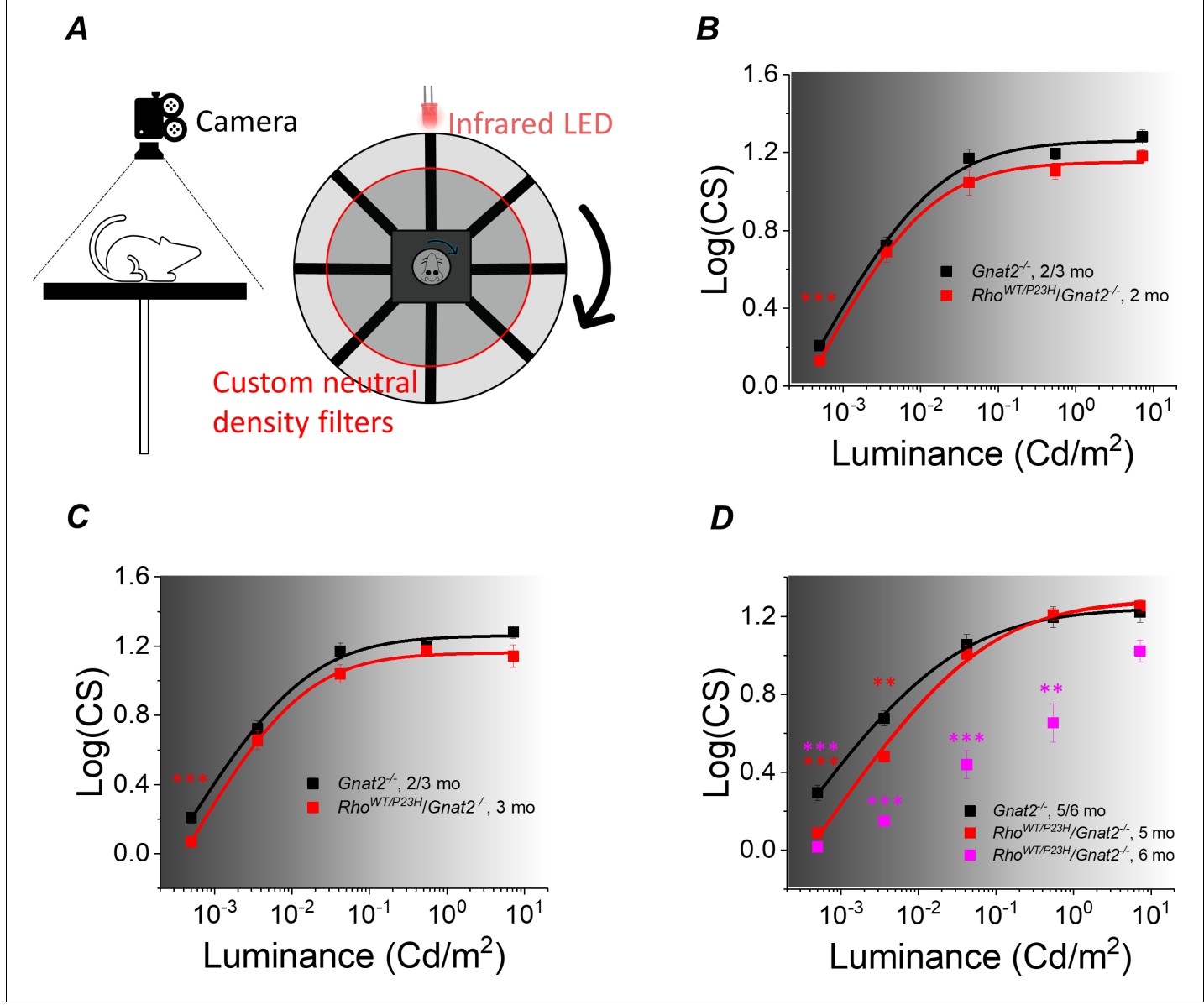

**Figure 7.** Visual contrast sensitivity (CS) mediated by rods is not severely compromised in P23H/$Gnat2^{-/-}$ mice up to 5 months of age. (**A**) Mouse contrast sensitivity threshold was tested using an optomotor reflex test and infrared visualization of the mouse head movement and cylindrical neutral density filters around the mouse platform. Contrast sensitivity was compared between $Rho^{WT/P23H}$/$Gnat2^{-/-}$ and their age-matched $Gnat2^{-/-}$ littermate mice from dim scotopic up to mesopic/photopic conditions at 1 (**B**), 3 (**C**), and 5/6 (**D**) month of age. Smooth lines plot Hill-type *Equation 3* fitted to mean data points. *p<0.05, **p<0.005, ***p<0.001, two-way ANOVA followed by Bonferroni's post-hoc test. (Control, 2–3 mo, n = 6 (24); P23H, two mo, n = 11(12); P23H, three mo, n = 6 (13); control, 5–6 mo, n = 9 (17); P23H, five mo, n = 4 (25); P23H, six mo, n = 3 (5) mice(experiments)).
The online version of this article includes the following source data and figure supplement(s) for figure 7:

**Source data 1.** Contrast sensitivity data from individual experiments measured from control ($Gnat2^{-/-}$) and P23H/$Gnat2^{-/-}$ mice underlying the graphical data presented B, C and D.
**Figure supplement 1.** Visual contrast sensitivity (CS) is not compromised in P23H/$C57$ mice up to 5 months of age.
**Figure supplement 1—source data 1.** SContrast sensitivity data from individual experiments measured from control (C57) and P23H mice.

distinct phases (*Jones and Marc, 2005*; *Pfeiffer et al., 2020*). In phase 1, glial cells are activated, and neural reprogramming begins due to photoreceptor death, typically starting in rods that mediate our night vision. Phase 2 refers to an advanced disease stage when collateral, secondary cone cell death ensues. Phase 3 is a terminal phase and is thought to begin when all the photoreceptors

have degenerated. According to this classification, our study using an adRP mouse model begins when remodeling is in phase 1 and ends in the earliest steps of phase 2 (*Figure 1*).

We employed knock-in mice that carry a heterozygous P23H rhodopsin mutation. The P23H mouse line is a well-characterized model for the most common form of adRP, and closely parallels the pathological characteristics of the human disease (*Sakami et al., 2014*; *Sakami et al., 2011*). Typically, the clinical disease first manifests as night blindness followed by a gradual loss of visual field (*Berson et al., 1991*). However, the disease severity and progression are moderate compared to many other types of RP, and the patients can retain normal visual acuity well into their thirties (*Berson et al., 1991*). This may be due to the cone population preserved in many patients (*Hartong et al., 2006*). We found that the number of cone cells remained normal in P23H mice up to 5 months of age, the last time point studied (*Figure 1K*). Cone function was normal in 1- and 3-month-old P23H mice but significantly decreased at 5 months of age when compared to their WT littermates (*Figure 1L*). An earlier study demonstrated that the maximal rod response amplitude is suppressed by only 5% at post-natal day (P) 14 in heterozygous P23H mice (*Sakami et al., 2014*), implying practically normal amplitudes at eye opening. We found that the maximal ERG response arising from the rod population was reduced by 45–60% in 1-month-old P23H mice compared to control mice depending on the method of recording (in vivo vs. ex vivo) and the background of the mice (C57 vs. $Gnat2^{-/-}$) (*Figures 1J*, *4E* and *5A*). However, the waveform of the ERG response appeared qualitatively normal, and the first positive deflection of the in vivo ERG signal (b-wave) (*Figure 1I*) as well as the rod bipolar cell responses derived using ex vivo ERG (*Figures 4F* and *5B*, *Table 1*) were comparable between 1-month-old P23H and control mice. Therefore, we concluded that 1 month of age represents an optimal time for a molecular genetics screen of P23H mice as by this age a functional deficit of the rod population is significant, whereas their degeneration is not yet severe (*Figures 1*, *2*, *4* and *5*). Furthermore, at 1 month of age the confounding factors arising from any developmental plasticity associated with the retina is avoided due to the full maturation of the mouse retina at this age (*Young, 1985*).

We first hypothesized that if homeostatic plasticity occurs in response to rod cell death in the retina, it must show a discernible fingerprint in the retinal transcriptome. As expected, due to a loss of rods, several photoreceptor selective gene clusters were downregulated in P23H retinas (*Figure 3B*, *Figure 3—figure supplement 2*). In contrast, in the early disease state pronounced upregulation arose in pathways involving neural development, such as synaptic organization, axonogenesis and cell adhesion (*Figure 3C,D*). We also found several enriched pathways that overlap between cell stress responses and neuronal plasticity, such as the MAPK, NF-kappa B and TNF signaling pathways (*Pozniak et al., 2014*). In the case of P23H retinas, it is difficult to determine whether these major transcription regulators are upregulated primarily due to cell degeneration and stress or due to the induction of plasticity machinery. Nevertheless, our RNA-seq analysis is consistent with our hypothesis that a robust neural network adaptation in the retina takes place in the early stages of photoreceptor degenerative disease caused by the P23H rhodopsin mutation. At the same 1 month of age, we found that while the photoreceptor component of the ex vivo ERG response was significantly suppressed (*Figures 4E* and *5A*), the light sensitivity of the rod bipolar cell responses was similar to that in control mice (*Figures 4F* and *5B*, *Table 1*). In the C57 background, the light-sensitivity of bipolar cells tended to be even higher in 1-month-old P23H mice as compared to WT controls (*Figure 4F*, *Table 1*). However, this may be due to the contribution from increased light responses of cones observed in vivo (*Figure 1L*), since without the contribution of cone phototransduction, the rod bipolar cell sensitivity slightly decreased in P23H/$Gnat2^{-/-}$ mice (*Figure 5B*, *Table 1*). We also found better preserved visually guided behavior on the C57 background (*Figure 7—figure supplement 1*) as compared to the $Gnat2^{-/-}$ background (*Figure 7*). However, the compensatory sensitization of the rod bipolar cells to their rod input cannot be explained simply by upregulation of cone-mediated signaling, since the same phenomenon was observed in P23H mice both in C57 and $Gnat2^{-/-}$ backgrounds (*Figure 5C*, *Table 1*). The P23H/$Gnat2^{-/-}$ mice were also observed to perform almost normally until 3 months of age in a behavioral visual contrast threshold task, even at the lowest luminance (*Figure 7*). At this age, the overall rod input already decreased by more than half in P23H mice which was expected to significantly elevate contrast thresholds (or decrease contrast sensitivity) specifically at the two lowest luminance levels where the contrast sensitivity is proportional to background light intensity (*Figures 5A* and *7*). Therefore, it appears that the gradual loss of the

rod photoreceptors, or declined input from them, triggers functional homeostatic adaptation in the retina that allows maintenance of visual function downstream of the photoreceptors.

## Is retinal remodeling and plasticity supporting or impairing retinal output and vision?

Our results using the P23H mouse model appear at odds with some prior research casting a rather negative connotation over retinal remodeling following the loss of photoreceptors. Such studies proposed that retinal remodeling exacerbates vision loss and therefore may interfere with strategies to restore vision in the blind (*Jones et al., 2003*; *Marc and Jones, 2003b*; *Telias et al., 2019*; *Toychiev et al., 2013*). What might underlie this apparent discrepancy with our findings? We speculate that the contrasting conclusions could stem from the retinal remodeling phase at which various retinal degeneration models with significantly different rates of photoreceptor degeneration have been studied. The list of retinal degeneration mouse models in which remodeling has been investigated is extensive (*Marc et al., 2003a*); however, currently the most utilized models according to the literature are *rd1* and *rd10* mice in which photoreceptors degenerate rapidly due to mutations of phosphodiesterase 6 (*Pde6*) in rods. The *rd1* mouse model represents an extremely severe disease where rod death starts at P8 and progresses to practically complete loss of all photoreceptors by 4 weeks of age. The *rd10* model exhibits somewhat slower progression of the disease. Their dark-adapted ERG response is observable but already significantly compromised at P18 and by 1 month of age the light responses of rods are practically abolished and most of their photoreceptors have degenerated, particularly in the central retina (*Chang et al., 2007b*; *Gargini et al., 2007*; *Wang et al., 2018*). Thus, it is possible that functional and behavioral studies using *rd1* and *rd10* mice, typically conducted with mice older than 1 month of age, may have missed the window for observing constructive remodeling expected to promote retinal output and vision. In contrast, the slower rate of retinal degeneration in the heterozygous P23H animal model provides a broader window of opportunity for investigating the impact of photoreceptor degeneration on early stage retinal remodeling (phase 1).

Recent studies using the specific deletion of ~50–90% of rods, cones or rod bipolar cells during development or in mature mice have revealed homeostatic mechanisms associated with the maintenance of normal retinal output and vision (*Care et al., 2020*; *Care et al., 2019*; *Johnson et al., 2017*; *Shen et al., 2020*; *Tien et al., 2017*). Here we used a P23H knock-in mouse model of RP that recapitulates the human disease. Our results are consistent with the idea that at least during phase 1 of remodeling, photoreceptor death induces functional compensation that supports close to normal retinal output and rod-mediated vision (*Figures 4*, *5* and *7*). Even at the stage where we observed prominent cone dysfunction (*Figure 1L*) in 5- or even 6-month-old P23H/*Gnat2*$^{-/-}$ mice, the rod bipolar cells continued to increase their sensitivity to rod input (*Figure 5C–F*, *Table 1*), although at this point the light responses of rods were already small (*Figure 5A*) and the absolute amplitudes of the rod bipolar cell light responses had significantly decreased (*Figure 5B*) together with lower contrast sensitivity of their rod-mediated vision in dim light (*Figure 7D*). These results therefore suggest that up to a point where some residual rod light responses exist, the retina can adapt toward supporting normal retinal output and scotopic vision. In this manner, patients at this stage of retinal degenerative disease might be expected to benefit from various treatments designed to restore photoreceptors and their light responses. However, the duration during which the inner retina remains capable of processing light signals and therefore is amenable to these types of treatments remains unknown. Studies with other animal models suggest that corruption of the inner retina would become a problem at later time points, including in P23H mutation-associated RP models. Quantitative functional and behavioral studies at later stages will be needed to address this question.

## Mechanism for increased sensitivity of rod bipolar cells to their rod input

Our RNA-seq and electrophysiology data strongly suggest that homeostatic plasticity is induced at early stages of rod degeneration in the P23H mouse retina. Still, the homeostatic plasticity mechanisms for the increased sensitivity of rod bipolar cells to their rod input remain unclear. Possibilities include (1) formation of new synapses between rod bipolar cells and surviving photoreceptors, (2) increased electrical coupling of rods to the cone pathway, (3) reduced inhibitory drive to rod bipolar

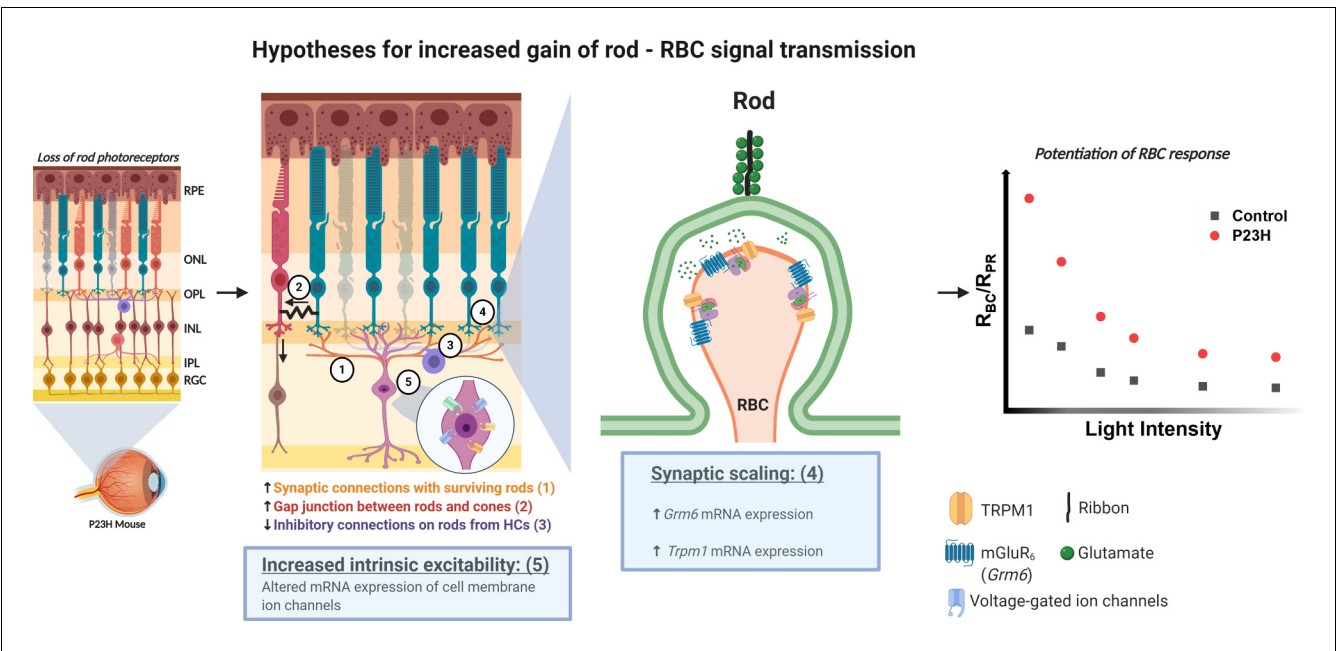

**Figure 8.** Hypotheses for the potentiation of the rod-rod bipolar cell signal transmission in the P23H mice. Numbers 1–5 indicate the five different hypotheses discussed in the text. Two of the most probable hypotheses highlighted with blue boxes. Created with BioRender.com.

cells (disinhibition), (4) pre- and/or postsynaptic scaling, and (5) increased excitability through axonal remodeling of rod bipolar cells (*Figure 8*). We tested hypotheses 2 and 3 by using blockers for gap junctions and major inhibitory postsynaptic receptors (*Figure 6*). Our findings did not support either of these hypotheses, and it appears that the potentiation of the ERG b-wave is not mediated by disinhibition or altered gap junctional coupling in P23H mice. Hypothesis 1 would align well with recent studies showing that cone bipolar cell dendrites formed new synapses with surviving cones after ~50% of cones had been ablated in juvenile or mature mice (*Care et al., 2019*; *Shen et al., 2020*). In another study, ablation of 50–90% of rod bipolar cells during development led to an extension of the dendrites of the remaining rod bipolar cells so that they contacted more rods (*Johnson et al., 2017*). It also has been shown that restoration of rod function in juvenile mice, which lacked rod function during development, can lead to the formation of new functional synapses between rods and rod bipolar cells (*Wang et al., 2019*). However, our data from IHC staining does not support formation of new synaptic contacts . We detected fewer mGluR6 positive puncta, reflecting a decreased number of synapses, in 1-month-old P23H retinas (*Figure 2*), and on the other hand, rod bipolar cell labeling showed indistinguishable staining patterns between P23H and control retinas (*Figure 1—figure supplement 1*). In line with this, ablation of ~50% of rods in a WT adult mouse retina, phenotypically mimicking the situation in young adult P23H mice including compensation of the ERG b-wave amplitude, did not lead to morphological synaptic changes in the study by *Care et al., 2020*. Our RNA-seq data show upregulations in a number of mRNAs of ion channels and G protein-coupled receptors, such as the *Trpm1* cation channel crucial for ON bipolar cell depolarization as well as glutamatergic receptors that could contribute either to the strengthening of individual rod-rod bipolar cell synapses or increased excitability of the rod bipolar cells (*Figure 3—figure supplements 3–5*). Therefore, postsynaptic scaling and/or increased intrinsic excitability of rod bipolar cells (hypotheses 4 and/or 5) appear the most likely candidates in explaining the functional compensation upon rod death in P23H mice (*Figure 8*). On the other hand, Care et al. proposed that functional compensation at rod bipolar cells upon ~50% input loss from rods was attributable to disinhibition (our hypothesis 3). It should be noted, however, that the mechanism by which neurons and neural networks adapt to acute stressors versus progressive disease conditions may differ. Another interesting possibility is that Müller cells would support bipolar cell function and

contribute to the preservation of b-wave response amplitudes in ex vivo ERG experiments. Although we used barium to block the slow PIII component (*Bolnick et al., 1979*), it is possible that Müller cells contribute to the ERG b-wave even in the presence of barium. If this contribution changes in P23H mice, that could in principle also explain the observed potentiation of the ex vivo ERG b-waves in the P23H mice. Additional work will be needed to resolve the exact homeostatic plasticity mechanisms that promote rod-rod bipolar cell signaling and visual function. Such knowledge could then help us understand how sensory systems respond to potentially compromising circumstances throughout life, and as well help in designing interventions to enhance homeostatic plasticity when needed.

# Materials and methods

## Key resources table

| Reagent type (species) or resource | Designation | Source or reference | Identifiers | Additional information |
|---|---|---|---|---|
| Genetic reagent *Mus musculus* | C57BL/6J | Jackson Laboratory | Stock #: 000664 RRID:MGI:5657312 | WT mouse |
| Genetic reagent *Mus musculus* | B6.129S6(Cg)-*Rho^{tm1.1Kpal}*/J | Jackson Laboratory | Stock #: 017628 RRID:IMSR_JAX:017628 | *Rho^{P23H/P23H}* mouse |
| Genetic reagent *Mus musculus* | Gnat2^{tm1(KOMP)Vlcg} | Marie E. Burns (University of California, Davis) | Original reference: doi: 10.1016/j.exer.2018.02.024 | *Gnat2^{-/-}* mouse |
| Antibody | Anti-PKCα (rabbit monoclonal) | Abcam | Cat. #: ab32376 RRID:AB_777294 | IHC (1:2000) |
| Antibody | Anti-Calbindin (mouse monoclonal) | Abcam | Cat. #: ab75524 RRID:AB_1310017 | IHC (1:1000) |
| Antibody | Anti-vGlut1 (guinea pig polyclonal) | Millipore | Cat. #: ab5905 RRID:AB_2301751 | IHC (1:1000) |
| Antibody | Anti-mGluR6 (sheep) | Jeannie Chen (University of Southern California) | Originally developed by Kirill Martemyanov lab, doi:10.1523/JNEUROSCI.1367–09.2009 | IHC (1:2000) |
| Antibody | Anti-CtBP2 (mouse monoclonal) | BD Biosciences | Cat. #: 612044 RRID:AB_399431 | IHC (1:1000) |
| Antibody | Anti-S opsin (goat polyclonal) | Krzysztof Palczewski lab | Generated by Bethyl Laboratories | IHC (1:1000) |
| Antibody | Anti-M opsin (rabbit polyclonal) | Novus biologicals | Cat. #: NB110-74730 RRID:AB_1049390 | IHC (1:1000) |
| Antibody | Anti-ERK1/2 (rabbit monoclonal) | Cell Signaling Technology | Cat. #: 4695S RRID:AB_390779 | WB (1:2000) |
| Antibody | Anti-α-tubulin (rabbit polyclonal) | Cell Signaling Technology | Cat. #: 2144S RRID:AB_2210548 | WB (1:2000) |
| Antibody | Anti-phospho ERK1/2 (mouse monoclonal) | Cell Signaling Technology | Cat. #: 9106S RRID:AB_331768 | WB (1:1000) |
| Antibody | Donkey anti-rabbit AlexaFluor 647 secondary | Abcam | Cat. #: ab150075 RRID:AB_2752244 | IHC (1:500) |

*Continued on next page*

*Continued*

| Reagent type (species) or resource | Designation | Source or reference | Identifiers | Additional information |
|---|---|---|---|---|
| Antibody | Donkey anti-mouse AlexaFluor 488 secondary | Abcam | Cat. #: ab150105 RRID:AB_2732856 | IHC (1:500) |
| Antibody | Donkey anti-mouse AlexaFluor 555 secondary | Abcam | Cat. #: ab150106 RRID:AB_2857373 | IHC (1:500) |
| Antibody | Goat anti-guinea pig AlexaFluor 568 secondary | Abcam | Cat. #: ab175714 | IHC (1:500) |
| Antibody | Donkey anti-sheep AlexaFluor 488 secondary | Abcam | Cat. #: ab150177 RRID:AB_2801320 | IHC (1:500) |
| Antibody | Donkey anti-goat AlexaFluor 488 secondary | Abcam | Cat. #: ab150129 RRID:AB_2687506 | IHC (1:500) |
| Antibody | IRDye 800CW goat anti-rabbit | Licor | Cat. #: 926–32211 RRID:AB_621843 | WB (1:10000) |
| Antibody | IRDye 680RD donkey anti-mouse | Licor | Cat. #: 926–68072 RRID:AB_10953628 | WB (1:10000) |
| Software, algorithm | Keyence BZ-X800 Analyzer | Keyence Corporation of America | RRID:SCR_017205 | Microscopy image analysis |
| Software, algorithm | MetaMorph 7.8 | Molecular Devices | RRID:SCR_002368 | Microscopy image analysis |
| Software, algorithm | GraphPad Prism 8 | GraphPad Software Inc | RRID:SCR_002798 | Statistical analysis |
| Software, algorithm | OriginPro 2020b | OriginLab Inc | https://www.originlab.com/ | Analysis and graphing software |
| Chemical compound, drug | CBR-5884 | Sigma Aldrich | Cat. #: 1656 | Phosphoglycerate dehydrogenase inhibitor |
| Chemical compound, drug | DL-AP4 | Tocris Bioscience | Cat. #: 0101/100 | Glutamatergic antagonist |
| Chemical compound, drug | Picrotoxin; PTX | Sigma Aldrich | Cat. #: 528105 | GABA antagonist |
| Chemical compound, drug | Strychnine; STR | Sigma Aldrich | Cat. #: S0532 | Glycine antagonist |
| Chemical compound, drug | Meclofenamic acid; MFA | Sigma Aldrich | Cat. #: M4531 | Gap junction blocker |
| Chemical compound, drug | (1,2,5,6-Tetrahydropyridin-4-yl)methylphosphinic acid; TPMPA | Sigma Aldrich | Cat. #: T200 | $GABA_A$-$\rho$ antagonist |
| Chemical compound, drug | L-741626 | Tocris Bioscience | Cat. #:1003 | $D_2$-antagonist |
| Chemical compound, drug | L-745870 | Tocris Bioscience | Cat. #:1002 | $D_4$-antagonist |

## Animals

Initially, we crossed $Rho^{P23H/P23H}$ males (*Sakami et al., 2011*) with C57Bl/6J female mice (The Jackson Laboratory, stock # 000664) to generate mice carrying the heterozygous P23H mutation in rhodopsin. Experimental ($Rho^{P23H/WT}$) and littermate control mice were generated using $Rho^{P23H/WT}$ and $Rho^{WT/WT}$ breeding pairs. We also generated a P23H mouse line on $Gnat2^{-/-}$ (*Ronning et al., 2018*) background by crossing P23H mice with $Gnat2^{-/-}$ mice, a kind gift from Dr. Marie Burns, University of California, Davis. Experimental ($RhoP^{23H/WT}/Gnat2^{-/-}$) and littermate controls ($Gnat2^{-/-}$) were generated using $Rho^{P23H/WT}/Gnat2^{-/-}$ and $Rho^{WT/WT}/Gnat2^{-/-}$ breeding pairs. Both male and female mice were used in this study. Mice were kept under 12/12 hr light cycle with free access to food and water. All experimental protocols adhered to Guide for the Care and Use of Laboratory Animals and were approved by the institutional Animal Studies Committees at the University of Utah and University of California, Irvine.

## Histology, immunohistochemistry, microscopy and image analysis

After euthanasia, the superior side of each mouse eye was marked with a thread burner or a permanent marker and then enucleated. Eyes for histology were fixed in Hartman's fixative (Sigma Aldrich, St. Louis, MO) for at least 24 hr. The eyes were kept in 70% ethanol before embedding into paraffin blocks. Sections were cut at 10 µm thickness in a nasal-temporal orientation. Retinal panorama images were captured using a light microscope. Representative magnified images shown in *Figure 1B,C* were taken at the superior middle retina centered at ~700 µm from the optic nerve head (ONH).

Eyes for cryosectioning and immunohistochemistry (IHC) were put in ice-cold 4% paraformaldehyde (PFA) immediately after enucleation, anterior parts of the eyes were removed, and eyecups separated. The eyecups were fixed for 15 min in ice-cold 4% PFA and then transferred into phosphate buffered saline (PBS) wash. Series incubation at 10, 20% and 30% sucrose in PBS was performed before embedding eyecups into cryoblocks using Tissue-Tek O.C.T. Compound (Sakura Finetek USA, Torrance, CA). Sections were cut at 10 µm thickness with a cryostat. Before IHC staining, the sections were thoroughly washed in PBS 4 × 15 mins. Blocking was performed first in 10% normal donkey serum in PBS, then in 10% normal goat serum, and finally in a Mouse on Mouse kit following the manufacturer's instructions (M.O.M. Immunodetection Kit, Basic, BMK-2202, Vector Laboratories, Burlingame, CA). Sections were incubated in primary antibody solutions (made in M.O. M. protein concentrate and 0.5% Triton X-100) at 4°C overnight on an orbital shaker. Next day, samples were washed 3 × 15 min in PBS+0.5% Triton X-100 (PBST), incubated in appropriate fluorescent secondary antibodies, washed again for 3 × 15 min, and finally mounted and secured on microscope slides with coverslips and Vectashield hardset mounting medium with DAPI (H-1500, Vector Laboratories). The primary antibodies used were rabbit anti-PKCα (ab32376, Abcam, Cambridge, UK, dilution 1:2000), mouse anti-calbindin (ab75524, Abcam, dilution 1:1000), guinea pig anti-vGlut1 (ab5905, Millipore, Burlington, MA, dilution 1:1000), sheep anti-mGluR6 (courtesy of Dr. Jeannie Chen, University of Southern California; dilution 1:2000), and mouse anti-CtBP2 (cat# 612044, BD Transduction Laboratories, San Jose, CA, dilution 1:1000); and secondary antibodies AlexaFluor's (Abcam) donkey anti-rabbit 647 nm, donkey anti-mouse 488 nm, goat anti-mouse 555 nm, goat anti-guinea pig 568 nm, and donkey anti-sheep 488 nm. All secondary antibodies were used at 1:500 dilution.

Fluorescence microscopy was performed using Keyence BZ-X800 microscope (Keyence Corp. USA, Itasca, IL) either using a 40x objective or a 100x oil immersion objective. All images were taken at a central location. Sixteen z-axis positions with 0.3 µm separation were taken at each x-y coordinate when using the 40x objective, and 50 z-axis positions with 0.2 µm separation when using the 100x objective to generate z-stacks. Maximum intensity projections of the z-stacks are displayed in figures. The exposure was kept at same level at each experimental condition, with the exception that PKCα signal was acquired at three times longer exposure when imaging 3-month and 5-month-old P23H samples compared to 1-month-old P23H and WT samples. Raw images were deconvoluted using Keyence BZ-X800 Analyzer's haze reduction tool retaining the same settings in all technical and biological replicates. Keyence BZ-X800 Analyzer's hybrid cell count tool was used to evaluate mGluR6 puncta count and average puncta size in 1-month-old P23H and WT samples. For this, five technical replicates per sample were acquired. These images were cropped to 100 µm width and 30

µm height as shown in *Figure 2C,D*. Puncta counts and sizes were averaged between the technical replicates within a sample for statistical analysis.

## Optical coherence tomography

Mice were anesthetized with ketamine (100 mg/kg, KetaVed, Bioniche Teoranta, Inverin Co, Galway, Ireland) and xylazine (10 mg/mg, Rompun, Bayer, Shawnee Mission, KS) by intraperitoneal injection, and their pupils were dilated with 1% tropicamide. OCT was performed with a Bioptigen spectral-domain OCT device (Leica Microsystems Inc, Buffalo Grove, IL). Four frames of OCT b-scan images were acquired from a series of 1200 a-scans. ONL thickness was measured 500 µm from the ONH at nasal, temporal, superior and inferior quadrants. The ONL thickness was averaged over the four retinal quadrants, and this average was used in the analysis.

## Preparation of retinal whole mounts and cone counting

Retina whole mounts for cone population analyses were performed as described earlier (*Leinonen et al., 2019*). In brief, the superior side of each eye was marked with a permanent marker before enucleation. Eyes were then fixed in 4% paraformaldehyde in isotonic phosphate buffered saline (PBS) for 1 hr after enucleation, and the retina was dissected away and processed as a whole-mount sample and subsequently stained using polyclonal goat S-opsin (1:1000 dilution, Bethyl Laboratories, Montgomery, TX) and polyclonal rabbit M-opsin (1:1000 dilution, Novus Biologicals, Littleton, CO) primary antibodies and fluorescent secondary antibodies (dilution for both 1:500; donkey anti-goat Alexa Fluor 488 and donkey anti-rabbit Alexa Fluor 647; Abcam). The whole retinal area was imaged using a fluorescence light-microscope (Leica DMI6000B, Leica Microsystems Inc) equipped with an automated stage, a 20 × objective, and green (excitation 480/40 nm) and far red (excitation 620/60 nm) fluorescence filter channels. Cone counts were computed from each individual image using MetaMorph 7.8 software (Molecular Devices, Sunnyvale, CA) and summed together to yield a whole retina count. For illustrative panorama images, the individual images were stitched together using MetaMorph 7.8 software.

## In vivo electroretinography (ERG)

In vivo ERG was performed as previously described (*Orban et al., 2018*) using a Diagnosys Celeris rodent ERG device (Diagnosys, Lowell, MA). Briefly, a mouse was anesthetized with ketamine and xylazine and placed on a heating pad at 37°C, and its pupils were moistened with 2.5% hypromellose eye lubricant (HUB Pharmaceuticals, Rancho Cucamonga, CA). Light stimulation was produced by an in-house scripted stimulation series in Espion software (version 6; Diagnosys). Before scotopic recordings, the mice were dark-adapted overnight and animal handling before recording was performed under dim red light. The eyes were stimulated with a green LED (peak 544 nm, bandwidth 160 nm) using a 13-step ascending flash intensity series ranging from 0.00015 to 300 cd·s/m$^2$.

Before photopic ERG recordings, mice were kept in a vivarium. After induction of anesthesia, the eyes were first adapted to a rod-suppressing green background light at 20 cd/m$^2$ for 1 min. Stimulation was performed with a blue LED (peak emission 460 nm, bandwidth ~100 nm) at light intensity increments of 0.1, 1.0, and 10.0 cd·s/m$^2$. As the short wavelength cone opsin (S-opsin) and medium wavelength cone opsin (M-opsin) sensitivities peak at 360 and 508 nm in mice (*Nikonov et al., 2006*), respectively, the blue LED stimulates both S- and M-opsins. LED light emission spectra were measured with a Specbos 1211UV spectroradiometer (JETI Technische Instrumente GmbH, Jena, Germany).

The ERG signal was acquired at 2 kHz and filtered with a low frequency cutoff at 0.25 Hz and a high frequency cutoff at 300 Hz. Espion software automatically detected the ERG a-wave (first negative ERG component) and b-wave (first positive ERG component).

## RNA isolation, quantitative PCR and RNA-sequencing

Mice were euthanized by cervical dislocation, and eyes were promptly harvested. Retinas were dissected away from the eyecup and put into RNAlater solution (Qiagen, Germantown, MD). The samples were stored at −80°C before processing. The retinas were homogenized using a motorized pestle, and the homogenate was spun in a tabletop centrifuge at full speed for 3 min. Supernatant was collected and RNA extracted using a Qiagen RNeasy Mini Kit following the manufacturer's

instructions. On-column DNAase digestion (Qiagen RNase-Free DNase) was used to remove any genomic DNA contamination from the sample. Five hundred ng of purified RNA were used for cDNA synthesis using an iScript cDNA synthesis kit (Bio-Rad, Irvine, CA), and real-time quantitative PCR (qPCR) was used as one of the initial sample quality checks before RNA-sequencing (RNA-seq), and also in verifying the expression level of *Grm6*, *Trpm1*, *Slc17a7* and *Cacna1F* using the ΔΔCt method. The combination of *Actb*, *Sdha* and *Gapdh* expression was used as a normalization control. The primer sequences are shown in *Table 2*.

Final RNA quality control (QC), library preparation, and RNA-seq were performed by Sacramento NovoGene Co., Ltd. (Sacramento, CA) using the Illumina HiSeq instrument. QC was performed with agarose gel electrophoresis and Nanodrop and Agilent 2100 devices. The gene expression data were aligned with a reference genome (*Mus musculus* mm10) using Tophat2 (http://ccb.jhu.edu/soft-ware/tophat/index.shtml). The gene expression level was measured by transcript abundance and reported as Fragments Per Kilobase of transcript sequence per Millions base pairs (FPKM) sequenced. As the first measure of dataset reliability, a Pearson correlation was conducted between all samples. Subsequently, to test data integrity between experimental groups (1-month WT female, n = 4; 1-month WT male, n = 3; 1-month P23H female, n = 4; 1-month P23H male, n = 3; 3-month WT, n = 3; 3-month P23H female, n = 2: 3-month P23H male, n = 2), the FPKM distribution was inspected in violin plots. Next, Venn diagrams were inspected to reveal relationships between the four groups, after which the sexes were pooled and only genotypes (WT females+males vs P23H females+males) were compared in further analyses. The Gene Ontology (GO) and KEGG analyses, and predicted protein-protein interaction analysis using the STRING database (https://string-db.org/cgi/input.pl), were obtained to reveal gene expression changes at the level of networks. Gene expression differences of single genes between genotypes was reported as $\log_2$(mean of P23H/mean of WT) and an adjusted p-value of <0.05 was used as a level of statistical significance.

Raw data is made freely available in the Gene Expression Omnibus (GEO) database (https://www.ncbi.nlm.nih.gov/geo/) with accession numbers GSE152474 (1-month-old samples) and GSE156533 (3-month-old samples).

## Immunoblotting

The mice were euthanized by cervical dislocation, eyes were promptly harvested, and retinas were quickly dissected away from the eyecup and immediately frozen using liquid nitrogen. A single retina extract was homogenized in ice-cold lysis buffer (70 µl) containing 100 mM Tris-HCl, 10 mM magnesium acetate, 6 M urea, 2% SDS, benzonase nuclease (25 U per ml), protease inhibitor (cOmplete Mini EDTA-free, Roche Diagnostics, Indianapolis, IN) and phosphatase inhibitor cocktails I and II (Sigma Aldrich). Homogenates were centrifuged and the supernatants mixed with Laemlli sample buffer (Bio-Rad). Protein concentration in each sample was determined using a Pierce BCA protein assay kit (ThermoFisher, Waltham, MA). Forty-eight µg of protein were loaded onto a Mini-Protean TGX gel (Bio-Rad) and electrophoresis was run on an ice-bed. Proteins were transferred to a nitrocellulose membrane. The membrane was blocked with 5% fat-free milk in Tris-buffered saline containing 0.1% Tween 20 (TBST) for 1 hr, and thereafter incubated in primary antibody in 1% milk in TBST. After an overnight incubation, the membrane was washed three times with TBST and thereafter incubated in fluorescent secondary antibody solution (LI-COR, Lincoln, NE) at 1:10,000 dilution for 2 hr. The membrane was washed three times with TBST and subsequently imaged using an Odyssey

**Table 2.** Primer sequences used in qPCR.

| Gene | Forward primer sequence | Reverse primer sequence |
| --- | --- | --- |
| *Actb* | GGCCAACCGTGAAAAGATGA | GACCAGAGGCATACAGGGAC |
| *Cacna1F* | CGGACGAATGCACAAGACAT | CGGTATGGTTCAGTGTGCAT |
| *Gapdh* | GACGGCCGCATCTTCTTG | CCAAATCCGTTCACACCGA |
| *Grm6* | CCATCACCATCTTGCCCAAA | CCAGAACTCAGCAAACCAGA |
| *Sdha* | GCAGTTTCGAGGCTTCTTC | CAACAGAGAAGTGAAAGCCG |
| *Slc17a7* | GGGTCCTTGTGCAGTATTCA | CAGTGCCGGTGACTCATAGG |
| *Trpm1* | CTGTCAGCAAACACACCCAG | GCCAGTCCTTCACCATGAG |

imager (LI-COR). The primary antibodies used were mouse anti-phospho-ERK1/2 (dilution 1:1000), rabbit anti-ERK1/2 (dilution 1:2000) and rabbit α-tubulin (dilution 1:2000) all purchased from Cell Signaling Technology (Danvers, MA). The phospho-ERK1/2 expression was evaluated first using a 680 nm secondary anti-mouse antibody, and the same membrane was re-incubated later in ERK1/2 and α-tubulin solution using an 800 nm secondary anti-rabbit antibody.

## Ex vivo electroretinogram

Transretinal (ex vivo) electroretinography (ERG) was conducted as described previously (*Vinberg et al., 2014*; *Vinberg et al., 2015a*). Briefly, mice were dark adapted for at least two hours and euthanized by $CO_2$ and cervical dislocation under dim red light. Eyes were enucleated and immediately placed into carbonated (95% $O_2$/5% $CO_2$) Ames' medium (A1420; Sigma-Aldrich, St. Louis, MO) containing 1.932 g/L $NaHCO_3$ (Sigma-Aldrich, St. Louis, MO). The retina was dissected from the eye and transferred to the recording chamber which was perfused at a rate of 1 mL/min with heated (37°C) Ames' medium containing 100 µM $BaCl_2$ (Sigma-Aldrich, St. Louis, MO) to remove the Müller glia contribution to the ERG signal (*Bolnick et al., 1979*). To isolate the photoreceptor component of the ERG signal, 40 µM DL-AP4 (Cat. #0101; Tocris Bioscience) was added to the perfusion medium. In some experiments (*Figure 6*), the following drugs were added into Ames' (µM; supplier Sigma-Aldrich, St. Louis, MO unless otherwise noted): 100 picrotoxin (PTX) to block GABA receptors, 10 strychnine (STR) to block glycine receptors, 100 meclofenamic acid (MFA) to block electrical coupling via gap junctions, 50 (1,2,5,6-Tetrahydropyridin-4-yl)methylphosphinic acid (TPMPA) to block GABA_C receptors, and 10 L-741626 and L-745870 (both from Tocris Bioscience) to block $D_2$ and $D_4$ receptors respectively. Drugs were first dissolved in ethanol (PTX), DMSO (L-741626) or water (STR, L-745870, MFA, TPMPA), and 0.02–0.2% of the stock solutions were added into Ames' to achieve the final concentration indicated above. For control solution, the same amount of water, DMSO or ethanol instead of the drug-containing stock solution was added.

From the dark-adapted retina, ERG responses to flashes of light (505 nm, 2–10 ms flash duration) from ~17 photons $\mu m^{-2}$ up to ~20,000 photons $\mu m^{-2}$ were recorded (10 trials/dimmer flash intensity and five trials/brighter flashes). ON bipolar cell responses were determined by subtracting the photoreceptor response (containing DL-AP4) from the ERG response without DL-AP4. Photoreceptor response and ON bipolar cell response amplitudes were measured and their ratio was calculated and plotted as a function of light flash intensity (see *Figure 4*). To determine flash intensity ($I$) required to generate half-maximal photoreceptor or ON bipolar cell responses ($I_{1/2}$), a Hill function

$$R = R_{max}\frac{I^n}{I_{1/2}^n + I^n},$$ (1)

was fitted to amplitude ($R$) data using the Originlab's (2018, build: 9.5.0193) nonlinear curve fit tool. $R_{max}$ is the maximal response amplitude to a bright flash (set based on data) and n is a steepness factor that was set to one for photoreceptor responses but was let to vary freely to produce optimal fits to $R_{BC}$ (bipolar cell response amplitude) data. We also fit similar Hill functions to data plotting $R_{BC}$ as a function of $R_{PR}$:

$$R_{BC} = R_{BC,max}\frac{R_{PR}^n}{R_{PR,1/2}^n + R_{PR}^n}$$ (2)

to determine the photoreceptor input required to generate half-maximal ON bipolar cell response, $R_{PR,1/2}$.

## Measurement of visual contrast sensitivity

An optomotor reflex test (OMR) was used to determine contrast sensitivity threshold using a commercially available OptoMotry system (CerebralMechanics Inc, Canada) (*Prusky et al., 2004*) customized with an infrared LED in the chamber for visualization of mouse head movement by a camera attached to the lid of the box (*Figure 7A*). Additional cylindrical neutral density filters (in layers) were placed around the mouse platform to assess contrast sensitivity in scotopic and mesopic-photopic conditions. The ambient luminance ($Cd/m^2$) was calibrated using an optometer (UDTi FlexOptometer, photometric sensor model 2151, Gamma Scientific, CA). Since the dynamic range of the sensor did not reach the dimmest conditions, those were derived based on separate measurements

of the attenuation factor for each neutral density filter layer. Before each experiment mice were dark-adapted overnight, and the testing started at the dimmest light condition. The test was administered in a single-blinded fashion, with the experimenter unaware of the stimulus presented to the mouse. Horizontally drifting vertical sine-wave gratings moving left or right with different contrast were presented at a spatial frequency of 0.128 cycles/degree and a grating rotation speed (drift speed) of 5.4 degrees/s to measure contrast thresholds (CT, 1–100%) at different ambient luminance ($L$) levels from $5*10^{-4}$ to 7 Cd/m$^2$. Contrast sensitivity (CS) is defined as 1/CT and was plotted as a function of ambient luminance. A Hill function defined by

$$CS = CS_{max} \frac{L^n}{L_{1/2}^n + L^n} \tag{3}$$

was fit to the mean luminance-CS data with $L_{1/2}$ (luminance at which $CS = 0.5\ C_{Smax}$), $CS_{max}$ (maximal CS at ~1–7 Cd/m$^2$) and n (steepness factor) as free parameters.

## Statistical analysis

ONL thickness analysis was tested using one-way ANOVA. Repeated measures two-way ANOVA was performed for ERG stimuli series comparisons. Two-way ANOVA was used to analyze OMR and $R_{BC}/R_{PR}$ data at various luminance/intensity levels. The ANOVAs were followed by Bonferroni's post hoc tests. Ex vivo ERG parameters from fitted functions or measured $R_{max}$ between control and P23H mice were analyzed by a two-tailed t-test. A paired two-tailed t-test was used to determine if inhibitors affected the ERG a- or b-waves (*Figure 6*). T-tests were corrected from multiple comparisons when analyzing mRNA expressions (*Figure 3—figure supplements 4–5*) using False Discovery Rate approach and two-stage step-up method of Benjamini, Krieger and Yekutieli in GraphPad Prism version eight software (San Diego, Ca). Results are displayed as mean ± SEM and statistical significance was set at p<0.05.

## Acknowledgements

HL was supported by Fight for Sight, Knights Templar Eye Foundation, Eye and Tissue Bank Foundation (Finland), the Finnish Cultural Foundation and the Orion Research Foundation. FV was supported by NIH grant EY026651, the International Retinal Research Foundation, and Research to Prevent Blindness/Dr. H James and Carole Free Career Development Awards. KP was supported by NIH grants EY009339 and R24 EY027283. KP is the Irving H Leopold Chair of Ophthalmology. The authors also acknowledge support from an RPB unrestricted grant to the Department of Ophthalmology, University of California, Irvine and to the Department of Ophthalmology and Visual Sciences, University of Utah. We thank Dr. Marie Burns (University of California, Davis) for providing the *Gnat2*$^{-/-}$ mouse strain used in this study, Dr. Jeannie Chen (University of Southern California) for mGluR6 antibody, and Dr. James Hall (University of California, Irvine) for proofreading the manuscript.

## Additional information

### Funding

| Funder | Grant reference number | Author |
| --- | --- | --- |
| National Eye Institute | R00 EY026651 | Frans Vinberg |
| International Retinal Research Foundation | Regular Grant | Frans Vinberg |
| Research to Prevent Blindness | Dr. H. James and Carole Free Career Development | Frans Vinberg |
| National Eye Institute | R01 EY009339 | Krzysztof Palczewski |
| National Eye Institute | R24 EY027283 | Krzysztof Palczewski |
| Eye and Tissue Bank Foundation | Postdoctoral Award | Henri Leinonen |

| Finnish Cultural Foundation | Postdoctoral Award | Henri Leinonen |
| Orion Research Foundation | Postdoctoral Award | Henri Leinonen |
| Research to Prevent Blindness | Unrestricted grant to the Department of Ophthalmology | Krzysztof Palczewski |
| Research to Prevent Blindness | Unrestricted grant to the Department of Ophthalmology and Visual Sciences | Frans Vinberg |
| Research to Prevent Blindness | University of Utah | Frans Vinberg |
| Research to Prevent Blindness | University of California Irvine | Krzysztof Palczewski |
| Knights Templar Eye Foundation | Career-Starter Research Grant | Henri Leinonen |
| Fight for Sight | Postdoctoral Award | Henri Leinonen |

The funders had no role in study design, data collection and interpretation, or the decision to submit the work for publication.

## Author contributions

Henri Leinonen, Conceptualization, Data curation, Formal analysis, Supervision, Funding acquisition, Validation, Investigation, Visualization, Methodology, Writing - original draft, Project administration, Writing - review and editing; Nguyen C Pham, Formal analysis, Investigation, Visualization, Methodology, Writing - review and editing; Taylor Boyd, Johanes Santoso, Investigation; Krzysztof Palczewski, Resources, Supervision, Funding acquisition, Writing - review and editing; Frans Vinberg, Conceptualization, Resources, Data curation, Formal analysis, Supervision, Funding acquisition, Validation, Investigation, Visualization, Methodology, Writing - original draft, Project administration, Writing - review and editing

## Author ORCIDs

Henri Leinonen https://orcid.org/0000-0002-0388-832X
Krzysztof Palczewski http://orcid.org/0000-0002-0788-545X
Frans Vinberg https://orcid.org/0000-0003-3439-4979

## Ethics

Animal experimentation: All experimental protocols adhered to Guide for the Care and Use of Laboratory Animals and were approved by the institutional Animal Studies Committees at the University of Utah (protocol #20-17015) and University of California, Irvine (protocol #AUP-18-124).

## Decision letter and Author response

Decision letter https://doi.org/10.7554/eLife.59422.sa1
Author response https://doi.org/10.7554/eLife.59422.sa2

# Additional files

## Supplementary files

• Supplementary file 1. Differentially expressed genes in P23H female versus P23H male mouse retinas at postnatal day 30.

• Supplementary file 2. Differentially expressed genes in WT female versus WT male mouse retinas at postnatal day 30.

• Supplementary file 3. Downregulated genes in P23H mouse retinas as compared to WT at postnatal day 30.

• Supplementary file 4. Upregulated genes in P23H mouse retinas as compared to WT at postnatal day 30.

- Supplementary file 5. Downregulated GO pathways in P23H mouse retinas as compared to WT at postnatal day 30.
- Supplementary file 6. Upregulated GO pathways in P23H mouse retinas as compared to WT at postnatal day 30.
- Supplementary file 7. Downregulated KEGG pathways in P23H mouse retinas as compared to WT at postnatal day 30.
- Supplementary file 8. Upregulated KEGG pathways in P23H mouse retinas as compared to WT at postnatal day 30.
- Supplementary file 9. Downregulated predicted protein-protein reactome pathways in P23H mouse retinas as compared to WT at postnatal day 30.
- Supplementary file 10. Upregulated predicted protein-protein reactome pathways in P23H mouse retinas as compared to WT at postnatal day 30.
- Supplementary file 11. Downregulated genes in P23H mouse retinas as compared to WT at postnatal day 90.
- Supplementary file 12. Upregulated genes in P23H mouse retinas as compared to WT at postnatal day 90.
- Supplementary file 13. Downregulated GO pathways in P23H mouse retinas as compared to WT at postnatal day 90.
- Supplementary file 14. Upregulated GO pathways in P23H mouse retinas as compared to WT at postnatal day 90.
- Supplementary file 15. Downpregulated KEGG pathways in P23H mouse retinas as compared to WT at postnatal day 90.
- Supplementary file 16. Upregulated KEGG pathways in P23H mouse retinas as compared to WT at postnatal day 90.
- Supplementary file 17. Downregulated predicted protein-protein reactome pathways in P23H mouse retinas as compared to WT at postnatal day 90.
- Supplementary file 18. Upregulated predicted protein-protein reactome pathways in P23H mouse retinas as compared to WT at postnatal day 90.
- Transparent reporting form

### Data availability

Sequencing data have been uploaded in GEO, accession numbers: GSE152474 (1-month-old samples) and GSE156533 (3-month-old samples).

The following datasets were generated:

| Author(s) | Year | Dataset title | Dataset URL | Database and Identifier |
|---|---|---|---|---|
| Leinonen H, Vinberg F | 2020 | Transcriptomic profiling in juvenile P23H Retinitis Pigmentosa mouse retinas | https://www.ncbi.nlm.nih.gov/geo/query/acc.cgi?acc=GSE152474 | NCBI Gene Expression Omnibus, GSE152474 |
| Leinonen H, Vinberg F | 2020 | Transcriptomic profiling in 3-month-old P23H Retinitis Pigmentosa mouse retinas | https://www.ncbi.nlm.nih.gov/geo/query/acc.cgi?acc=GSE156533 | NCBI Gene Expression Omnibus, GSE156533 |

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
