## [Decision Letter]

**Acceptance summary:**

The paper investigated retinal remodeling and its underlying mechanism in P23H mice with retinal degeneration. The authors demonstrated that strengthening synapse formation occurred as a response to rod loss and rod-rod bipolar signaling maintained night vision in P23H mice. This finding suggests a potential therapeutic approach by targeting retinal adaption as a common route to maintain retinal function despite various genetic causes of retinal degeneration.

**Decision letter after peer review:**

Thank you for submitting your article "Homeostatic plasticity in the retina is associated with maintenance of night vision during retinal degenerative disease" for consideration by *eLife*. Your article has been reviewed by three peer reviewers, including Lois Smith as the Reviewing Editor and Reviewer #1, and the evaluation has been overseen by Chris Baker as the Senior Editor. The following individual involved in review of your submission has agreed to reveal their identity: Wallace B Thoreson (Reviewer #3).

The reviewers have discussed the reviews with one another and the Reviewing Editor has drafted this decision to help you prepare a revised submission.

Summary:

The paper investigated retinal remodeling and its underlying mechanism in P23H mice with retinal degeneration. It fills a current knowledge gap of retinal adaption during retinal degeneration: adaptive responses in bipolar cell dendrites during retinal degeneration have been shown in recent studies but such the impact of these changes is unknown. With a longitudinal study of retinal function using ERG and the mechanistic exploration using RNA-seq, the authors demonstrated that increased synapse formation occurred as a response to rod loss and rod-rod bipolar signaling maintained night vision in P23H mice. This finding suggests a potential therapeutic approach (for future studies) by targeting retinal adaption as a common route to maintain retinal function despite various genetic causes of retinal degeneration.

Essential revisions:

The major revisions suggested with the understanding that with the pandemic this may not be possible at present but the manuscript be revised to either limit claims to those supported by data in hand, or to explicitly state that the relevant conclusions require additional supporting data.

1) Changes in bulk RNA-seq profile in the degenerating retina will at least in part reflect changes in the cellular composition (i.e., loss of rods). Doing some IHC experiments could help support the conclusion that homeostatic synaptic rewiring occurs. This conclusion is important for the significance of the paper.

2) BaCl2 is an effective way to remove Muller cell influences on the ERG. In Figure 3, would 100uM BaCl2 completely remove the signal components arising from Muller glia? How did the authors determine the contribution of Muller glia to the b wave? Did the authors examine the ex vivo retina ERG without BaCl2 treatment?

It would also be important to include the discussion of potential impacts of Muller glia on maintaining visual function in P23H mice.

---

## [Author Response]

Summary:The paper investigated retinal remodeling and its underlying mechanism in P23H mice with retinal degeneration. It fills a current knowledge gap of retinal adaption during retinal degeneration: adaptive responses in bipolar cell dendrites during retinal degeneration have been shown in recent studies but such the impact of these changes is unknown. With a longitudinal study of retinal function using ERG and the mechanistic exploration using RNA-seq, the authors demonstrated that increased synapse formation occurred as a response to rod loss and rod-rod bipolar signaling maintained night vision in P23H mice. This finding suggests a potential therapeutic approach (for future studies) by targeting retinal adaption as a common route to maintain retinal function despite various genetic causes of retinal degeneration.

We agree with this statement. This work should facilitate future research to refine the molecular mechanism of the retinal adaptation to photoreceptor death and to aid in designing therapeutic strategies to support vision during degeneration. However, we would like to point out that our results did not demonstrate increased synapse formation (on the contrary we saw a modest decrease in synapses, please see above and below) but rather suggests strengthening of remaining, individual synapses. We emphasized this hypothesis in the revised manuscript, and softened down narration about dendritic remodeling in the Abstract and Discussion as it could be easily interpreted to mean structural rather than functional remodeling.

Essential revisions:The major revisions suggested with the understanding that with the pandemic this may not be possible at present but the manuscript be revised to either limit claims to those supported by data in hand, or to explicitly state that the relevant conclusions require additional supporting data.1) Changes in bulk RNA-seq profile in the degenerating retina will at least in part reflect changes in the cellular composition (i.e., loss of rods). Doing some IHC experiments could help support the conclusion that homeostatic synaptic rewiring occurs. This conclusion is important for the significance of the paper.

We agree with reviewers. We certainly observed a lot of transcriptomics changes primarily due to a mass loss of rods, and considered investigation of the downregulated pathways somewhat pointless in our study. A future single-cell RNA-seq study, perhaps published as an *eLife* research advance, should address this gap. However, bulk sequencing is able to pick up postsynaptic changes with less disturbance caused by rod loss. For instance, genes such as *Grm6* and *Trpm1*, essential for ON bipolar cell depolarization, were upregulated in 1-month-old P23H retinas as measured by FPKMs from RNA-seq, and later confirmed by qPCR (see Figure 3—figure supplements 4, 5). Importantly, this upregulation remained as disease advanced from early (1-mo) to a more intermediate state (3-mo). The current manuscript aimed at testing if functional network adaptation takes place upon progressive photoreceptor degeneration in a robust and easily understandable paradigm accessible to a wide readership. We believe this goal was achieved by using a simple transcriptomics screen.

However, for the revised manuscript we added IHC experiments to label rod terminals using antibody against the vesicular glutamate transporter 1 (VGLUT1), bipolar cells using antibody for protein kinase (PKC)-α and horizontal cells using antibody for Calbindin (see Figure 1—figure supplement 1). These results show that at one month of age the overall morphology of the bipolar or horizontal cells is not altered in P23H mice based on standard fluorescence light microscopy. The staining pattern of PKCα was indistinguishable at the OPL in 1-month WT and P23H samples, and it seems unlikely that bipolar cells undergo clear-cut “expansion”. Later at 3- and 5-month P23H retinas, the intensity of PKCα decreased and horizontal cells appear to degenerate. As expected due to rod degeneration, VGLUT1 staining decreased with age in the P23H mouse retinas. These results do not suggest formation of new synapses in line with a recent paper from Care et al., 2020, where ablation of rods led to a functional potentiation without corresponding anatomical changes. The PKCα/Calbindin/vGLUT1 staining, however, did not accurately assess details of synaptic densities and counts at the OPL. To attempt this, we co-stained pre- and postsynaptic terminals with antibodies for a ribbon synapse marker CtBP2 and mGluR6, respectively, and performed imaging at high resolution using optical sectioning. These data revealed, as expected, localization of the mGluR6 and CtBP2 puncta adjacent to each other both in WT and P23H mouse retinas (new Figure 2). We quantified the number and size of the mGluR6 puncta in several different sites in control and P23H mouse retinas and found that the count was modestly decreased in P23H samples, which is likely attributable to overall thinning of OPL. The mGluR6 puncta size, and overall density and appearance of synapses were unchanged in P23H samples (see 300x images in Figure 2E, F).

To summarize, our data do not indicate formation of new synaptic contacts as a mechanism of synaptic adaptation. Rather, we believe that remaining (post)synapses gain strength, as shown by Care et al., 2020, with a photoreceptor ablation model. Our gene expression data point in same direction. These hypotheses are now emphasized in the revised Discussion.

2) BaCl2 is an effective way to remove Muller cell influences on the ERG. In Figure 3, would 100uM BaCl2 completely remove the signal components arising from Muller glia? How did the authors determine the contribution of Muller glia to the b wave? Did the authors examine the ex vivo retina ERG without BaCl2 treatment?It would also be important to include the discussion of potential impacts of Muller glia on maintaining visual function in P23H mice.

Very interesting point as it is well-known that Müller glia cells respond early in many retinal degenerative diseases. The slow PIII component, originating in Müller cells, overlaps significantly with the b-wave and diminishes the observed bipolar cell component of the ex vivo ERG signal in a way that does not necessarily reflect true activity of the bipolar cells. In typical ex vivo ERG experiments, the electrical component from Müller glia is removed completely by the use of 100 μM BaCl_2_ (see Figure 2D in Vinberg and Kefalov, 2015) and, thus, our experiments are expected to reflect the adaptation of retinal neurons. It is worth noting, though, that Müller cells were not chronically blocked during our follow-up in P23H mice. In fact, even during the experiments Barium would only have blocked their barium-sensitive K^+^ channels but not necessarily other functions which could play a role in supporting bipolar cell function.

To address reviewers comment about potential contribution of Müller cell component to the ERG b-waves, we conducted additional experiments by comparing ex vivo ERG b-waves in the absence and presence of 100 μM Barium in 1-month *Gnat2^-/-^* and *P23H/Gnat2^-/-^* mice (Figure 5—figure supplement 1). As expected, b-wave amplitudes in the absence of Barium were smaller than in the presence of Barium, probably due to a negative overlapping of the slow PIII component both in WT and P23H mice. As we found with Barium, the b wave amplitudes without Barium were also quite similar between 1-month control and P23H mice (Figure 5—figure supplement 1D, E). We also isolated the slow PIII by subtraction (see Figure 5—figure supplement 1A-C) and found that it was not affected by P23H mutation in 1-month mice in dim light but appeared somewhat smaller with brighter flashes (Figure 5—figure supplement 1F). Consequently, our results do not support a hypothesis that increased Müller cell activity could explain the observed potentiation of the b-waves in P23H mice. One potential confounding factor remains, however, as it is possible that 100 μM Barium failed to block some other unknown positive component generated in part by activity of Müller cells. Potentiation of this component could also in principle explain the observed potentiation of the b-wave amplitudes, but we think this explanation is highly unlikely. However, discovering unknown positive components, and their potential modulation by RP disease, would be an extremely interesting finding, and we briefly discuss this possibility in the Discussion.